# SHOULD WE STILL PRETRAIN ENCODERS WITH MASKED LANGUAGE MODELING?

**Hippolyte Gisserot-Boukhlef**[1,5*]    **Nicolas Boizard**[2,5*]
**Manuel Faysse**[5]    **Duarte M. Alves**[6,7]    **Emmanuel Malherbe**[1]    **André F. T. Martins**[3,6,7]
**Céline Hudelot**[5]    **Pierre Colombo**[4]

[1]Artefact Research Center    [2]Diabolocom    [3]TransPerfect    [4]Cohere
[5]MICS, CentraleSupélec, Université Paris-Saclay    [6]Instituto de Telecomunicações
[7] Instituto Superior Técnico & Universidade de Lisboa (Lisbon ELLIS Unit)
`hippolyte.gisserot-boukhlef@centralesupelec.fr`

## ABSTRACT

Learning high-quality text representations is fundamental to a wide range of NLP tasks. While encoder pretraining has traditionally relied on Masked Language Modeling (MLM), recent evidence suggests that decoder models pretrained with Causal Language Modeling (CLM) can be effectively repurposed as encoders, often surpassing traditional encoders on text representation benchmarks. However, it remains unclear whether these gains reflect an inherent advantage of the CLM approach or arise from confounding factors such as model and data scale. In this paper, we address this question through a series of large-scale, carefully controlled pretraining ablations, training a total of 38 models ranging from 210 million to 1 billion parameters, and conducting over 15,000 fine-tuning and evaluation runs. We find that while training with MLM generally yields better performance across text representation tasks, CLM-trained models are more data-efficient and demonstrate improved fine-tuning stability. Building on these findings, we experimentally show that a biphasic training strategy that sequentially applies CLM and then MLM achieves optimal performance under a fixed computational training budget. Moreover, we demonstrate that this strategy becomes more appealing when initializing from readily available pretrained CLM models, reducing the computational burden needed to train best-in-class encoder models. We release all project artifacts at `https://hf.co/MLMvsCLM` to foster further research.

## 1 INTRODUCTION

Learning meaningful representations is essential for a wide range of natural language processing (NLP) tasks, such as sequence classification, named entity recognition, extractive question answering, and information retrieval. Traditionally, these representations are learned with encoders pretrained solely with Masked Language Modeling (MLM) (Devlin et al., 2019). More recently, however, decoder models pretrained with Causal Language Modeling (CLM) and later adapted with MLM have challenged the MLM-only paradigm (BehnamGhader et al., 2024), achieving state-of-the-art results on the Massive Text Embedding Benchmark (MTEB) (Muennighoff et al., 2023; Enevoldsen et al., 2025). Yet, these results have so far only been observed on models that are significantly larger than typical encoders and trained on substantially more data (Lee et al., 2024; Meng et al., 2024; Kim et al., 2024; Muennighoff et al., 2024). As a result, it remains unclear whether the success of this training paradigm stems from CLM, or merely from increased scale.

In this paper, we present a controlled study examining the impact of training with MLM and CLM on learning textual representations. We compare models with equal architecture and size, trained on

---

[*]Equal contribution

[1]`https://hf.co/MLMvsCLM`

[2]`https://github.com/Nicolas-BZRD/EuroBERT/tree/MLM_vs_CLM`

[3]`https://github.com/artefactory/EncodEval/tree/MLM_vs_CLM`

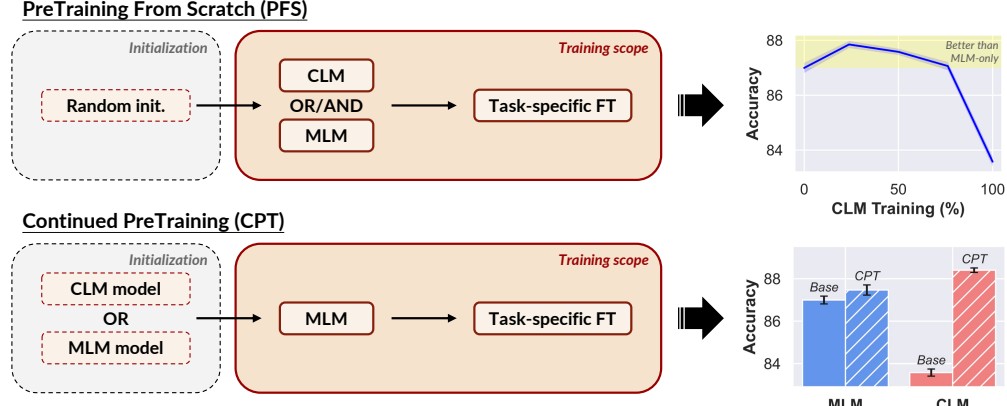

Figure 1: Experimental setup overview and key results on sequence classification (610M model size, 40% MLM ratio). The upper plot shows SC downstream performance as a function of the CLM-to-MLM step ratio during pretraining. The lower plot illustrates the effect of applying MLM CPT to models initially trained with either MLM or CLM (Base).

an equal amount of data, and evaluate them on an extensive range of text representation tasks. We start by comparing pretraining exclusively with either approach, and then investigate a two-stage pretraining alternative, where CLM and MLM are applied sequentially. Additionally, leveraging existing pretrained models, we investigate a continued pretraining setup, where training starts from models initially trained with either MLM or CLM and is then extended with MLM for a portion of the pretraining steps. In total, we train 38 models ranging from 210 million to 1 billion parameters and conduct over 15,000 fine-tuning and evaluation runs to ensure fair design comparisons, amounting to 110k MI250X GPU hours. These large-scale experiments show that:

- Although CLM training delivers strong performance on certain tasks and demonstrates good data efficiency and fine-tuning stability, MLM remains essential for achieving robust performance across all tasks, underscoring that bidirectional training remains essential (section 3).

- When pretraining a model from scratch, a two-stage protocol that first applies CLM followed by MLM offers a data-efficient way to build strong text representations, effectively combining the strengths of both paradigms (section 4).

- In a continued pretraining setting, adapting a CLM-pretrained model with MLM proves more effective than continuing MLM training from an MLM-pretrained model. This result suggests that leveraging widely pretrained decoder models is currently the best approach to obtain a strong encoder model (section 5).

To facilitate reproducibility and support future research, we release all pretrained model check-points,[1] together with the corresponding training[2] and evaluation codebases.[3]

## 2 EXPERIMENTAL SETUP

In this section, we describe our controlled experimental setup, covering model architectures, pretraining and fine-tuning protocols, as summarized in Figure 1.

**Models.** The model architectures closely follow those of the EuroBERT models (Boizard et al., 2025), with sizes of 210M, 610M, and 1B parameters. All models use a maximum context length of 2,048 tokens and a RoPE $\theta$ value of 10,000.[4]

**Pretraining data.** Models are trained on unique English tokens from the FineWeb-Edu dataset (Penedo et al., 2024), which is known for supporting efficient model training. To ensure fair com-

---

[4]Additional details on the model architectures can be found in Appendix A.

parison across model configurations and training setups, all models are exposed to the same sequence of samples during training.

**Pretraining paradigms.** Models are trained using one of 3 approaches:

1. **CLM** employs next-token prediction, where each token is generated autoregressively using a causal attention mask. Given an input sequence $\mathbf{x} = (x_1, x_2, \ldots, x_T)$, the training objective is to minimize the negative log-likelihood:

$$\mathcal{L}_{\text{CLM}}(\mathbf{x}) = -\sum_{t=1}^{T} \log p_{\theta_\rightarrow}(x_t \mid x_1, \ldots, x_{t-1}), \tag{1}$$

where $p_{\theta_\rightarrow}(\cdot \mid \cdot)$ denotes the model's predicted distribution over the vocabulary conditioned on the preceding tokens under the causal attention mask.

2. **MLM** randomly masks a subset of tokens and trains the model to reconstruct them using a bidirectional attention mask. The sequence-level training objective is given by:

$$\mathcal{L}_{\text{MLM}}(\mathbf{x}) = -\sum_{i \in \mathcal{M}} \log p_{\theta_\leftrightarrow}(x_i \mid \mathbf{x}_{\mathcal{M}}), \tag{2}$$

where $p_{\theta_\leftrightarrow}(\cdot \mid \cdot)$ denotes the model's predicted distribution under a bidirectional attention mask, $\mathcal{M} \subset \{1, \ldots, T\}$ specifies the indices of the masked tokens, and $\mathbf{x}_{\mathcal{M}}$ is the input sequence with the corresponding tokens replaced by special placeholders.[5] In practice, tokens are masked uniformly and independently, each with probability $p_{\text{mask}}$. In our experiments, we consider $p_{\text{mask}} \in \{20\%, 30\%, 40\%, 50\%\}$.

3. Finally, **CLM+MLM** sequentially combines the former approaches in a two-stage setup, where CLM pretraining is performed first, followed by MLM.

**Pretraining hyperparameters.** Pretraining is performed with a per-device batch size of 12 samples across 192 GPUs, yielding an effective batch size of 2,373,120 tokens.[6] We employ a Warmup-Stable-Decay (WSD) learning rate schedule: a 2,000-step warmup phase, followed by 38,000 steps with a constant learning rate of $5 \times 10^{-4}$,[7] ending with a 2,000-step linear decay phase, for a total of 42,000 training steps.

**Pretraining setups.** We consider two pretraining setups reflecting common real-world scenarios: training a model from scratch and continuing training from an existing model.

1. **Pretraining From Scratch (PFS)**. Models are trained from random initialization for a fixed number of steps using one of 3 approaches: CLM, MLM, or sequential CLM+MLM. For CLM and MLM, a standard WSD learning rate scheduler is applied. For CLM+MLM models, training is first performed with CLM, and then resumed with MLM from CLM checkpoints that have not undergone learning rate decay.

2. **Continued PreTraining (CPT)**. Training is initialized from models pretrained using either CLM or MLM and is then continued with MLM. In contrast to the PFS setup, the pretrained models used for CPT have already undergone learning rate decay during their initial training phase, reflecting real-world constraints and continued training practices. Another key difference is that CPT starts from checkpoints where the loss has already converged, whereas in PFS, the paradigm switch typically occurs while gradient norms are still large and learning is active.

**Fine-tuning tasks and datasets.** We evaluate all models across a broad range of text representation tasks, focusing on 4 key categories commonly used in real-world applications. For Sequence Classification (SC), we use SST-2 (Socher et al., 2013), MNLI (Williams et al., 2018), and QQP (Wang et al., 2017). For Token Classification (TC), we evaluate on the English subsets of CoNLL (Tjong Kim Sang & De Meulder, 2003), OntoNotes (Hovy et al., 2006), and UNER (Mayhew et al.,

---

[5]For example, if $T = 5$ and $\mathcal{M} = \{1, 3\}$, then $\mathbf{x}_{\mathcal{M}} = ([\text{mask}], x_2, [\text{mask}], x_4, x_5)$.

[6]Inputs consist of variable-length sequences ranging from 12 tokens up to the maximum of 2,048.

[7]The optimal learning rate was selected from the range $1 \times 10^{-4}$ to $2 \times 10^{-3}$ based on experiments using 10% of the training data.

2024). Question Answering (QA) is assessed using SQuAD (Rajpurkar et al., 2016), SQuAD-v2 (Rajpurkar et al., 2018), and ReCoRD (Wang et al., 2019). For Information Retrieval (IR), we use MS MARCO (Bajaj et al., 2016), NQ (Kwiatkowski et al., 2019), and the English subset of MLDR (Chen et al., 2024) for long-context evaluation.[8]

**Fine-tuning protocol.** To ensure a fair comparison, all models are fine-tuned using a consistent protocol. Each model is trained for up to 1,000 steps or one full epoch, whichever comes first, using a batch size of 32. To account for differences in model architecture and task requirements, we perform a grid search over 6 learning rates ($1 \times 10^{-5}, 2 \times 10^{-5}, 5 \times 10^{-5}, 1 \times 10^{-4}, 2 \times 10^{-4}$, and $5 \times 10^{-4}$) for each model-dataset pair, with 10% warmup followed by linear decay. The learning rate yielding the best validation performance is selected. Fine-tuning employs bidirectional attention mask with task-specific loss functions: for SC, we use cross-entropy on mean-pooled token embeddings; TC and QA are trained using token-level cross-entropy; and for IR, we rely on the InfoNCE loss (Oord et al., 2018) with in-batch negatives, using mean pooling. To account for the fine-tuning instability commonly observed in BERT-style models for representation learning (Devlin et al., 2019; Lee et al., 2020; Dodge et al., 2020; Zhang et al., 2020), the entire procedure is repeated across 5 random seeds. Fine-tuning is conducted on the in-domain training set, except for NQ and MLDR, for which training is performed on MS MARCO.

**Evaluation metrics.** SC is assessed with accuracy, TC and QA with F1 score, and IR with NDCG@10. We report results averaged across seeds, along with 95% confidence intervals.

**Infrastructure.** All pretraining and fine-tuning experiments are carried out on MI250X GPUs provided by the Adastra supercomputer, using the AMD-optimized EuroBERT codebase (Boizard et al., 2025). Pretraining runs are executed on 24 nodes with 8 GPUs each (192 GPUs total), while fine-tuning is carried out on single GPUs.

**Experimental scale.** Drawing reliable conclusions about model design choices typically requires scaling up and repeating experiments, as statistically sound results often require a large number of runs and sufficient training to reach minimal convergence (Hoffmann et al., 2022). We therefore carefully design our experiments to ensure robust and well-supported findings. The default pre-training setup uses 100B tokens, which corresponds to 5 times the optimal data budget proposed by Hoffmann et al. (2022) for decoder models in the 1B-parameter range. The total compute budget amounts to 110k GPU hours, broken down as follows:

- **Base pretraining** includes all 3 model sizes (210M, 610M, and 1B), trained with both CLM and MLM. For MLM, 4 masking ratios (20%, 30%, 40%, and 50%) are explored, resulting in a total of 15 models, each trained for 42,000 steps (100B tokens), and accounting for 81k GPU hours.

- **PFS experiments** explore various CLM-to-MLM step ratios (0%-100%, 25%-75%, 50%-50%, 75%-25%, and 100%-0%) across 3 training lengths (12,000, 22,000, and 42,000 steps), focusing on the 610M model size with a 40% masking ratio for MLM. This setup is also used to evaluate the effect of masking ratio on models pretrained with CLM followed by MLM. To minimize redundant computation, we reuse checkpoints from the base pretraining runs whenever possible, applying learning rate decay only during the final 2,000 steps. In total, these runs result in 17 new models and account for 120,000 additional training steps and 16k GPU hours.

- **CPT experiments** are performed on 610M models with a 40% masking ratio and varying training lengths (2,000, 12,000, and 22,000 steps) applied to both CLM- and MLM-pretrained models. To avoid redundant computation, training is resumed from shared checkpoints whenever possible. These runs yield 6 new models, totaling 26,000 training steps and 4k GPU hours.

- **Model evaluation** involves fine-tuning 50 model checkpoints in total, including 38 final pretrained checkpoints and 12 intermediate ones. Each model is fine-tuned on 12 datasets, across 6 learning rates and 5 random seeds, resulting in 15,120 fine-tuning runs which total about 9k GPU hours. Allocating substantial compute to the evaluation phase is particularly important in representation learning, where fine-tuning can exhibit high variance and instability (Devlin et al., 2019; Lee et al., 2020; Dodge et al., 2020; Zhang et al., 2020).

---

[8]Further details are provided in Appendix B.

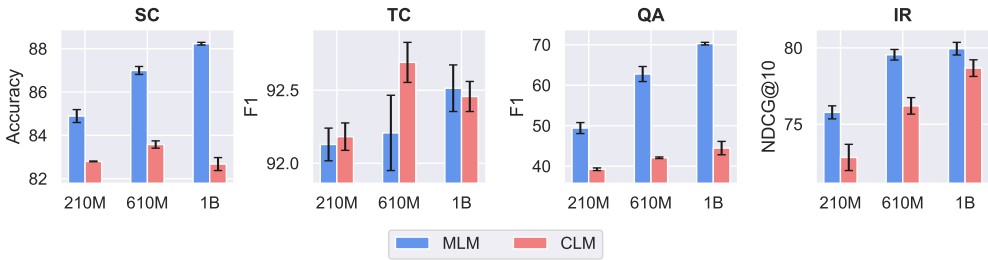

Figure 2: Downstream performance of MLM and CLM pretraining, averaged across tasks and reported for all model sizes. For MLM, results are given for a 40% masking ratio.

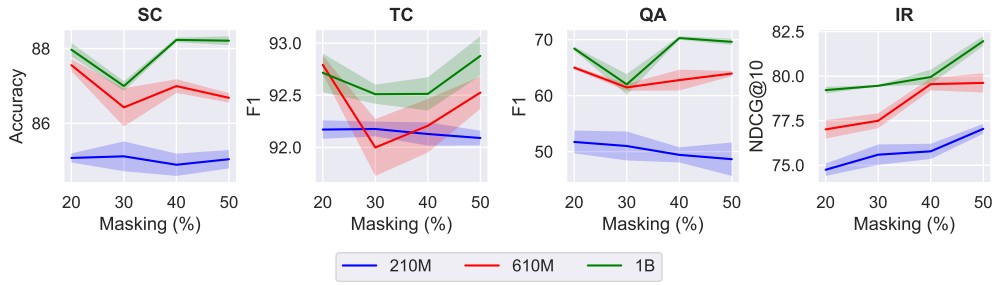

Figure 3: MLM downstream performance across different masking ratios for all model sizes.

# 3 PRETRAINING WITH CLM OR MLM

This section presents preliminary results comparing MLM and CLM pretraining in terms of downstream performance, data efficiency, and training and fine-tuning stability.

**MLM generally outperforms CLM on text representation tasks.** Bidirectional attention during pretraining tends to enhance representation quality on downstream tasks (Figure 2). The performance gap between models trained with MLM and those trained with CLM is especially noticeable on SC and QA tasks, and remains consistent across model sizes. The largest discrepancy is observed on QA, which appears particularly sensitive to the absence of bidirectional attention during pretraining. Interestingly, some task-specific trends emerge: for instance, the MLM-to-CLM gap widens with increasing model size on SC, but narrows on IR.

**There is no universally optimal masking ratio.** The masking ratio is a key design choice when pretraining models with MLM. As shown in Figure 3, there is no single best ratio, as it depends on both model size and downstream task, making MLM pretraining a delicate balance. Larger models tend to benefit from higher masking ratios, consistent with prior findings from Wettig et al. (2023). Across tasks, IR datasets consistently prefer higher masking ratios regardless of model size. In contrast, for token-level tasks such as TC and QA, smaller models perform better with lower masking ratios. For larger models (610M and 1B), the performance curves exhibit a U-shape, indicating improved performance at both low and high masking ratios.[9][10] To reduce computational overhead, subsequent experiments report results on 610M models trained with a 40% masking ratio for MLM, which provides a strong overall compromise across tasks.

**CLM models can perform competitively.** Interestingly, although CLM-pretrained models typically underperform on SC and QA tasks, they achieve solid results on IR, with the gap to MLM

---

[9]Additional results in Appendix C further highlight the influence of masking ratio, showing substantial variation even among datasets within the same task category.

[10]In some configurations, we observe smaller models outperforming larger ones on average (e.g., the 210M model surpassing the 610M model on TC). Such behavior can arise from inherent pretraining variance (Grattafiori et al., 2024; Yang et al., 2025) and, in this case, results in no statistically significant differences.

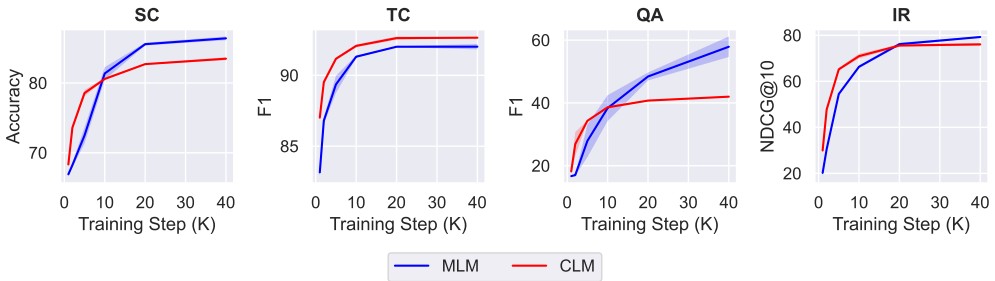

Figure 4: Downstream performance with respect to pretraining steps for CLM and MLM. Results are reported for 610M models, with a 40% masking ratio for MLM.

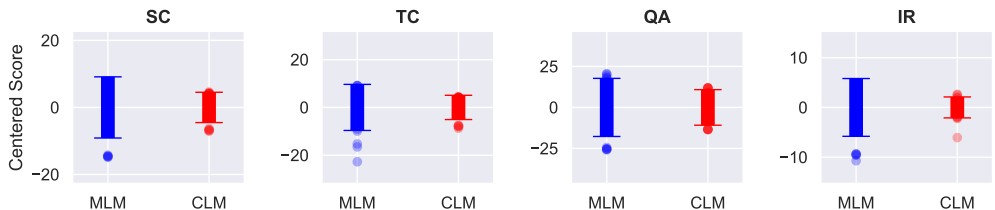

Figure 5: Impact of the fine-tuning learning rate on MLM- vs. CLM-pretrained models. Error bars indicate the standard deviation of metric scores across all seeds and learning rates between $10^{-5}$ and $10^{-4}$. Results are shown for 610M-parameter models, with a 40% masking ratio for MLM.

models narrowing as model size increases. On TC, they consistently match MLM models and even outperform them by a substantial margin at the 610M scale (Figure 2).[11] This suggests that causal pretraining can produce strong token-level representations, highlighting its potential despite the absence of bidirectional context.

**CLM is more data-efficient than MLM in the early stages of training.** As shown in Figure 4, CLM consistently outperforms MLM in downstream performance during the early stages of training (up to step 10,000 for SC and QA, 20,000 for IR, and even until the end for TC). However, as training progresses, MLM models tend to catch up and often surpass CLM, while CLM shows more limited gains in later steps. This suggests that although CLM saturates earlier, it enables more efficient representation learning in fewer steps. Notably, this makes CLM an appealing option for data-scarce scenarios, such as pretraining on low-resource languages, or simply as a warmup stage before MLM-based encoder training.

**CLM-based pretraining improves fine-tuning stability.** A key challenge in fine-tuning models for text representation tasks is selecting optimal hyperparameters, particularly the learning rate, which can be sensitive to factors like model size, task type, or dataset scale, making exhaustive grid searches computationally expensive. As shown in Figure 5, models pretrained with CLM demonstrate lower sensitivity to learning rate choices than those pretrained with MLM. This indicates that CLM pretraining provides a more stable initialization for fine-tuning, facilitating more reliable performance and reducing the need for extensive hyperparameter tuning.

## 4 TWO-STAGE CLM+MLM PRETRAINING

Building on section 3, which highlights clear advantages of CLM over MLM in terms of token-level representation quality, data efficiency, and training stability, we investigate the benefits of applying CLM and MLM sequentially during pretraining. We focus on the PFS setting, where a

---

[11]Intuitively, this behavior may stem from the nature of token-level tasks, which rely less on bidirectionally informed context to identify named entities or word types than SC, IR, or QA, which depend more heavily on broader contextual understanding. This hypothesis could be validated through further research.

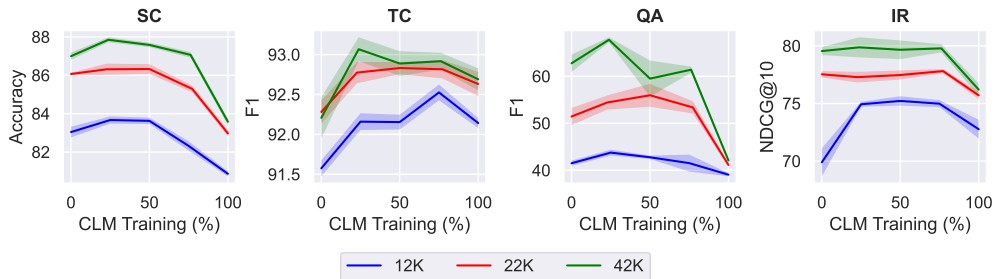

Figure 6: Impact of two-stage CLM+MLM pretraining on downstream performance under different training budgets (12,000, 22,000, and 42,000 steps). The x-axis shows the percentage of CLM steps allocated in the first phase. Experiments are conducted on 610M models, with a 40% masking ratio during MLM training.

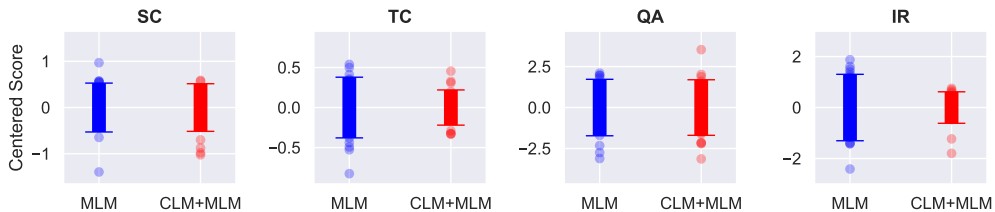

Figure 7: Comparison of downstream performance variability across different masking ratios (20%, 30%, 40%, 50%) for CLM and CLM+MLM pretraining configurations. Error bars indicate the standard deviation across fine-tuning seeds and masking ratios. Results are reported for 610M models, using 42,000 training steps for MLM and a schedule of 40,000 CLM steps followed by 2,000 MLM steps for CLM+MLM.

single learning rate scheduler is used, and evaluate this approach under fixed compute budgets of 12,000, 22,000, and 42,000 training steps, using different splits: 100% CLM, 25%-75%, 50%-50%, 75%-25%, and 100% MLM. To keep computation costs manageable, we focus on the 610M model scale and use a fixed masking ratio of 40%.

**Under fixed compute constraints, starting pretraining with CLM and continuing with MLM yields better results than pure MLM.** As shown in Figure 6, combining CLM and MLM consistently improves downstream performance compared to MLM-only training. In particular, the 25%-75% split reliably surpasses the MLM baseline, while even a 75% CLM allocation maintains comparable results. These findings confirm the benefits of CLM pretraining for learning strong text representations and showcase the synergies it provides when combined sequentially with MLM training.

**CLM-based models exhibit lower sensitivity to masking ratio.** As shown in Figure 7, adapted CLM models show less variation in performance across different masking ratios compared to fully MLM-trained models. Initial CLM pretraining appears to stabilize model performance, making adaptation more robust to masking ratio choices.

## 5 CONTINUED PRETRAINING FROM CLM AND MLM MODELS

In this section, we focus on the CPT setting, starting from equally sized CLM and MLM models pretrained on the same data (100B tokens in total). The key question is whether it is more beneficial to allocate additional compute to applying MLM CPT to a CLM model or to further train an MLM model. We consider 3 CPT budgets: 2,000, 12,000, and 22,000 steps, and focus on the 610M model scale with a 40% masking ratio to keep computational costs manageable.

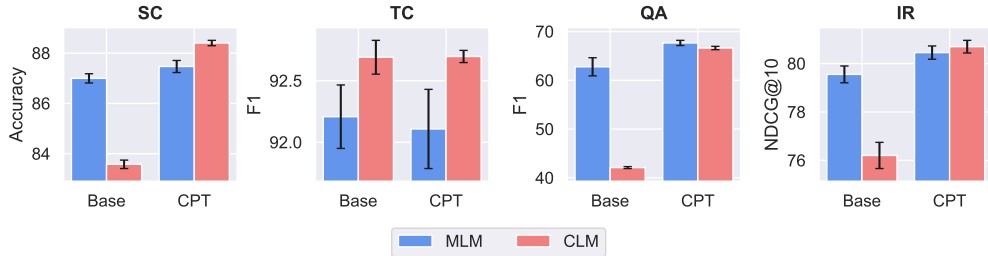

Figure 8: Impact of performing CPT on either CLM- or MLM-pretrained models (denoted as Base). CPT is conducted with MLM for 22,000 steps on 610M models with a 40% masking ratio, following 42,000 steps of initial pretraining.

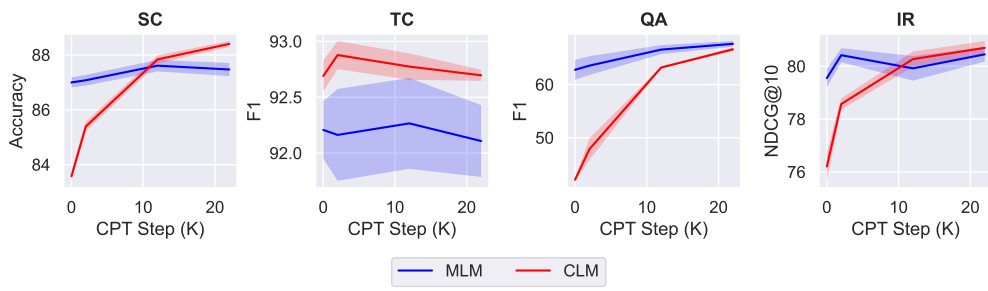

Figure 9: Downstream performance as a function of CPT length for CLM- and MLM-pretrained models, reported for settings of 2,000, 12,000, and 22,000 training steps. Experiments use 610M models with a 40% MLM ratio.

**MLM CPT on a CLM-pretrained model outperforms MLM-only training.** As shown in Figure 8, the MLM-adapted CLM model consistently achieves superior downstream performance. On TC, where CLM-only models were already strong, performance is maintained and the gap to MLM remains. For QA and IR, the gap is effectively closed, while for SC, the MLM-adapted CLM model significantly outperforms the MLM-only model.

**Fewer CPT steps already show strong performance.** Interestingly, we observe that it is not necessary to run as many as 22,000 CPT steps to achieve comparable performance between MLM and adapted CLM. As early as 12,000 steps, the results are already strong and broadly match those of MLM-only CPT (Figure 9), with better results on TC and IR, comparable performance on SC, and nearly on par on QA. In terms of trends, applying MLM CPT on a CLM model also appears more promising, showing a steeper improvement curve toward the end, while MLM-only training seems to plateau (particularly noticeable on SC).

## 6 RELATED WORK

Learning universal text representations has been a central focus in NLP, with MLM emerging as the dominant pretraining paradigm following the success of models like BERT (Devlin et al., 2019). MLM-based encoders leverage bidirectional attention to capture rich contextual dependencies, enabling strong performance across a wide range of tasks such as classification, question answering, and named entity recognition. Subsequent work has explored architectural extensions (e.g., RoBERTa (Liu et al., 2019), DeBERTa (He et al., 2021), ModernBERT (Warner et al., 2024b)) and more efficient pretraining strategies (Clark et al., 2020). In contrast, decoder-only architectures such as GPT-style models were originally optimized for autoregressive generation (Brown et al., 2020; Grattafiori et al., 2024; Jiang et al., 2023; Almazrouei et al., 2023), making them less naturally suited for text representation tasks.

Nevertheless, numerous studies have successfully explored adapting decoder-only models (typically larger and trained on more data than standard encoders) for learning high-quality text representations (Muennighoff, 2022; Lee et al., 2024; Meng et al., 2024; Zhang et al., 2025; Lee et al., 2025). As a result, these adapted decoders now rank among the top-performing models on prominent embedding benchmarks such as MTEB (Muennighoff et al., 2023; Enevoldsen et al., 2025). However, their strong performance does not disentangle the underlying factors (likely a combination of model scale and data regime) notably leaving open the question of whether causal pretraining itself plays a meaningful role in downstream representation quality.

Several recent studies have investigated the key factors influencing the performance of text representation models. On the MLM-only side, Wettig et al. (2023) explore the impact of the masking ratio on downstream tasks such as sequence classification and question answering, with a focus on learning dynamics, model scale, and fine-tuning behavior. From another perspective, BehnamGhader et al. (2024) propose a general framework for adapting decoder-only models to embedding tasks, along with a series of ablations designed to isolate the primary contributors to performance. Complementary work explores ways to reintroduce bidirectional attention into causal decoder models, such as by repeating sequences (Springer et al., 2024), using custom-shaped attention masks (Raffel et al., 2023), or selectively removing the causal mask on portions of the sequence in a post-training phase (Kopiczko et al., 2024; Beyer et al., 2024). Articles most closely related to ours propose controlled setups for comparing CLM, MLM, and hybrid strategies, but have either focused on very small amounts of pretraining data (Charpentier & Samuel, 2024), or prioritized other design parameters. For example, Weller et al. (2025) train on very large data regimes (2T tokens), analyze MLM-then-CLM learning curricula, and evaluate on generative benchmarks, providing practical insights into the impact of training design on downstream performance. However, their approach does not support extensive evaluation on encoder-specific tasks and limits exploration of additional training design choices, such as masking ratio or the CLM-MLM share in pretraining. Building on these efforts, we provide a comprehensive analysis of how the choice of learning objective and attention mask during training affects downstream representation quality. By employing a unified and controlled pretraining setup and evaluating across a broad range of representation tasks with statistically grounded results, we ensure fair comparisons and reduce confounding factors, enabling more precise attribution of observed effects.

# 7 CONCLUSION

Our study challenges the long-standing assumption that Masked Language Modeling (MLM) is the universally optimal approach for encoder pretraining. Through large-scale experiments at the 100B training token scale, we provide robust empirical evidence that encoders can be trained more data-efficiently without relying solely on MLM. In particular, we demonstrate that decoder-only models trained with Causal Language Modeling (CLM) offer clear advantages in data efficiency, training stability, and performance on specific text representation tasks. Moreover, we show that sequential training (first with CLM followed by MLM) is more effective than MLM alone. These results suggest that the most effective path to high-performing encoder models may involve leveraging pretrained decoder models and continuing with MLM-based training, paving the way for future state-of-the-art encoders at a discounted price. We hope our findings and released resources will support further advancements in this area.

# 8 LIMITATIONS AND FUTURE WORK

Our study deliberately focuses on a targeted set of parameters, namely learning objective, model size, training setup, data budget, and masking ratio. As in large-scale studies such as (Kaplan et al., 2020; Hoffmann et al., 2022), this required fixing several other parameters, including the model architecture, tokenizer, language, and training mixture. While these factors are certainly of interest, varying them would introduce significant computational overhead, making them difficult to incorporate into our current scope. We therefore view them as a promising direction for future work, including extensions to multilingual settings or to additional modalities such as vision-language.

Additionally, although we provide insights into scaling behaviors with respect to both model size and data budget, we constrained these dimensions to 1B parameters and 100B training tokens. This

allowed us to keep compute costs manageable while still exploring a broad range of design parameters. While 100B tokens correspond to roughly five times the compute-optimal budget suggested by Hoffmann et al. (2022) for 1B-parameter decoder models, and most pre-trained encoders in the literature remain below the 1B scale (Devlin et al., 2019; Liu et al., 2019; He et al., 2021; Warner et al., 2024a), we still view pushing beyond these limits as an important direction for future work, as many top entries in the MTEB benchmark (Muennighoff et al., 2023; Enevoldsen et al., 2025) frequently exceed the 1B-parameter range (Zhang et al., 2025; Babakhin et al., 2025). Exploring more sophisticated training curricula also appears promising for improving downstream performance, for instance, alternating CLM and MLM objectives multiple times throughout pretraining.

In a similar spirit, our study focuses on the extensive evaluation of pre-trained models and deliberately does not include contrastive post-training for zero-shot retrieval. Applying such post-training across a large number of training configurations would have incurred substantial computational costs and could have introduced additional confounding factors that require careful control (Xiong et al., 2020; Qu et al., 2021; Wang et al., 2022; Mohr et al., 2024; Chen et al., 2024; Sturua et al., 2024). While this lies outside the scope of the present work, exploring contrastive post-training would be a valuable way to complement and extend our findings.

Finally, several observations emerged throughout the paper that appear intriguing but fall outside the current scope of our study. Future work could, for example, further investigate the relationship between masking ratio and downstream performance across tasks. This includes providing deeper insights into patterns such as the U-shaped curve observed in Figure 3 for TC on the 610M and 1B models, in contrast with the monotonic trend observed for IR. More broadly, gaining a deeper understanding of how pretraining learning objectives transfer to different downstream tasks remains an interesting yet largely unexplored direction with substantial promise.

## ETHICS STATEMENT

**Environmental and compute considerations.** This work examines different pretraining strategies for encoder models, with the aim of exploring approaches that may reduce computational costs while still producing useful text representations. By considering alternatives to traditional Masked Language Modeling (MLM) and investigating a biphasic pretraining strategy, we seek ways to train encoder models in a more data- and compute-efficient manner. Reducing the computational resources required for model training can have both environmental and practical benefits. More efficient encoders may help lower energy consumption and the associated footprint of large-scale NLP training. Additionally, effective encoder models could in some cases reduce the need to rely on large generative models for downstream tasks, thereby decreasing dependence on more compute-intensive systems.

**Responsible use of LLMs.** In preparing this manuscript, we occasionally used suggestions from LLMs (GPT-5) to guide improvements in clarity, grammar, and overall readability. All scientific content, including experimental design, codebase, data analysis, results, and interpretations, is independently developed by the authors. LLMs are not involved in generating, modifying, or interpreting any experimental results, nor in producing code or analyses. Their use is strictly limited to selectively refining language to ensure clear and effective communication of our research.

## REPRODUCIBILITY STATEMENT

We have made every effort to ensure that our experiments are reproducible. All pretraining and evaluation experiments are described in detail in the paper, including model architectures, training objectives, data sources, and hyperparameter settings. To facilitate replication, we release all project artifacts, including pretrained models, training scripts, and evaluation code.

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

## A  TRAINING SETUP DETAILS

This appendix provides additional information on the training parameters used in this study. Table 1 outlines both the architectural configurations and training settings.

| Model Size | 210M | 610M | 1B |
|---|---|---|---|
| **Architecture** | | | |
| Layers | 12 | 26 | 28 |
| Embedding Dimension | 768 | 1,152 | 1,728 |
| FFN Dimension | 3,072 | 4,096 | 5,120 |
| Attention Heads | 12 | 18 | 18 |
| Key/Value Heads | 12 | 6 | 6 |
| Layer Normalization | RMSNorm | | |
| RMSNorm $\epsilon$ | $1 \times 10^{-5}$ | | |
| Activation Function | SwiGLU | | |
| Maximum Context Length | 2,048 | | |
| Positional Embeddings | RoPE | | |
| RoPE $\theta$ | 10,000 | | |
| Tokenizer | LLaMA 3 | | |
| Vocabulary Size | 128,256 | | |
| **Training** | | | |
| Weight Initialisation | $\mathcal{N}(\mu = 0, \sigma^2 = 0.2)$ | | |
| Learning Rate | $5 \times 10^{-4}$ | | |
| Scheduler | WSD (linear decay) | | |
| Warmup Steps | 2,000 | | |
| Decay Steps | 5% | | |
| Optimizer | AdamW | | |
| Adam $\beta_1$ | 0.9 | | |
| Adam $\beta_2$ | 0.95 | | |
| Adam $\epsilon$ | $1 \times 10^{-5}$ | | |
| Weight Decay | 0.1 | | |
| Gradient Clipping Norm | 1.0 | | |
| Per-GPU Batch Size | 12 | | |
| Gradient Accumulation Steps | 1 | | |
| GPUs | 192 | | |
| Tokens/Step | 2,359,296 | | |

Table 1: Architecture and training settings for the 3 model sizes under consideration.

## B  DETAILS ON EVALUATION DATASETS

This appendix offers additional details on the datasets used for evaluation.

**SC datasets:**

All SC datasets are sourced from the GLUE (Wang et al., 2018) benchmark.

- **MNLI** (Williams et al., 2018) — MNLI is a large-scale benchmark for natural language inference, where each example pairs a premise with a hypothesis, and the task is to determine whether the hypothesis is entailed by, contradicts, or is neutral with respect to the premise. Since the test set is not publicly available, we report results on the validation set and reserve 5% of the training data for validation. We evaluate using the matched subset, which reflects in-domain performance.

- **QQP** (Wang et al., 2017) — Binary classification dataset from Quora that asks whether a pair of questions are semantically equivalent. Similarly to MNLI, we report results on the validation set and reserve 5% of the training data for validation.

- **SST-2** (Socher et al., 2013) — Sentiment classification dataset consisting of movie review sentences. Each sentence is labeled as expressing either a positive or negative sentiment. We report results on the validation set and reserve 5% of the training data for validation.

**TC datasets:**

- **CoNLL** (Tjong Kim Sang & De Meulder, 2003) — NER dataset comprising English and German newswire text, annotated with 4 entity types: person, organization, location, and miscellaneous. In this work, we use only the English portion.
- **OntoNotes** (Hovy et al., 2006) — A large-scale dataset for NER and other NLP tasks, spanning diverse genres including newswire, broadcast news, and conversational speech in both English and Chinese. For this study, we focus on the English portion and evaluate on named entity spans.
- **UNER** (Mayhew et al., 2024) — Multilingual NER dataset based on Wikipedia and Wikidata annotations. For this work, we use only the English subset.

**QA datasets:**

- **ReCoRD** (Wang et al., 2019) — Challenging extractive QA dataset that tests commonsense reasoning by requiring models to select entities from a passage to answer cloze-style questions. Since the test set is not publicly available, we report results on the validation set and reserve 5% of the training data for validation.
- **SQuAD** (Rajpurkar et al., 2016) — Benchmark for extractive question answering. It consists of questions posed on Wikipedia articles, with answers given as text spans within the corresponding passages. We report results on the validation set and reserve 5% of the training data for validation.
- **SQuAD-v2** (Rajpurkar et al., 2018) — Extension of SQuAD that includes both answerable and unanswerable questions. We report results on the validation set and reserve 5% of the training data for validation.

**IR datasets:**

- **MLDR** (Chen et al., 2024) — Multilingual retrieval benchmark focused on long documents. In this work, we use only the English subset. Since MLDR does not provide training data, models are evaluated directly on the validation and test splits without task-specific fine-tuning.
- **MS MARCO** (Bajaj et al., 2016) — Large-scale English passage retrieval dataset derived from real Bing search queries and web documents. MS MARCO is used in this work for both training and evaluation.
- **NQ** (Kwiatkowski et al., 2019) — Open-domain English retrieval dataset based on real Google search queries, where answers are typically found in Wikipedia articles. As for MLDR, models are evaluated directly on the validation and test splits without task-specific fine-tuning.

To reduce computational overhead during evaluation on retrieval datasets, we restrict the corpus to labeled documents only, that is, those marked as positive or negative. Unlabeled documents are excluded from the evaluation.

## C  DETAILED RESULTS

### C.1  MLM VS. CLM

This appendix provides detailed results comparing CLM and MLM pretraining, including variations in masking ratios and model sizes. Table 2, Table 3, Table 4, and Table 5 present dataset-level results for SC, TC, QA, and IR, respectively.

### C.2  DATA EFFICIENCY

This appendix provides detailed results on data efficiency for MLM vs CLM models at different pretraining steps. Results are presented for 610M models with 40% masking for MLM. Table 6, Table 7, Table 8, and Table 9 present dataset-level results for SC, TC, QA, and IR, respectively.

| Model Size | Objective | Masking | SST-2 | QQP | MNLI | Average |
|---|---|---|---|---|---|---|
| 210M | MLM | 20% | 89.93 (±0.26) | **86.00** (±0.18) | 79.29 (±0.32) | 85.07 (±0.12) |
| | | 30% | 89.84 (±1.24) | 85.82 (±0.36) | **79.69** (±0.22) | **85.12** (±0.40) |
| | | 40% | 90.44 (±0.71) | 85.52 (±0.21) | 78.72 (±0.20) | 84.89 (±0.30) |
| | | 50% | **91.61** (±0.35) | 85.52 (±0.49) | 78.00 (±0.51) | 85.04 (±0.25) |
| | CLM | - | 90.00 (±0.30) | 83.84 (±0.27) | 74.56 (±0.16) | 82.80 (±0.02) |
| 610M | MLM | 20% | **92.61** (±0.57) | 86.98 (±0.06) | **83.08** (±0.33) | **87.56** (±0.16) |
| | | 30% | 91.49 (±0.69) | 86.48 (±0.25) | 81.31 (±1.27) | 86.43 (±0.51) |
| | | 40% | 91.42 (±0.62) | 86.98 (±0.15) | 82.59 (±0.16) | 87.00 (±0.18) |
| | | 50% | 91.93 (±0.26) | 86.38 (±0.34) | 81.74 (±0.36) | 86.68 (±0.13) |
| | CLM | - | 91.10 (±0.39) | 84.14 (±0.12) | 75.49 (±0.22) | 83.58 (±0.17) |
| 1B | MLM | 20% | 92.39 (±0.47) | 87.20 (±0.18) | 84.33 (±0.13) | 87.97 (±0.18) |
| | | 30% | 92.13 (±0.56) | 86.46 (±0.20) | 82.40 (±0.24) | 87.00 (±0.13) |
| | | 40% | 92.89 (±0.22) | **87.41** (±0.17) | **84.40** (±0.25) | **88.23** (±0.06) |
| | | 50% | **93.28** (±0.25) | 87.27 (±0.17) | 84.09 (±0.16) | 88.21 (±0.12) |
| | CLM | - | 92.48 (±0.46) | 83.66 (±0.11) | 71.89 (±0.44) | 82.68 (±0.30) |

Table 2: MLM vs. CLM downstream performance results on SC datasets.

| Model Size | Objective | Masking | OntoNotes | CoNLL | UNER | Average |
|---|---|---|---|---|---|---|
| 210M | MLM | 20% | 92.68 (±0.06) | 91.38 (±0.15) | 92.46 (±0.10) | 92.17 (±0.09) |
| | | 30% | **92.71** (±0.14) | 91.37 (±0.19) | 92.45 (±0.13) | **92.18** (±0.07) |
| | | 40% | 92.61 (±0.09) | **91.70** (±0.19) | 92.08 (±0.28) | 92.13 (±0.11) |
| | | 50% | 92.46 (±0.12) | 91.56 (±0.10) | 92.25 (±0.16) | 92.09 (±0.07) |
| | CLM | - | 92.18 (±0.07) | 91.67 (±0.32) | **92.70** (±0.10) | **92.18** (±0.09) |
| 610M | MLM | 20% | 92.93 (±0.13) | **92.34** (±0.17) | 93.10 (±0.29) | **92.79** (±0.09) |
| | | 30% | 92.69 (±0.15) | 91.37 (±0.40) | 91.94 (±0.64) | 92.00 (±0.27) |
| | | 40% | **92.97** (±0.15) | 91.58 (±0.44) | 92.07 (±0.77) | 92.21 (±0.26) |
| | | 50% | 92.95 (±0.20) | 91.79 (±0.16) | 92.84 (±0.33) | 92.53 (±0.15) |
| | CLM | - | 92.79 (±0.05) | 92.16 (±0.21) | **93.12** (±0.29) | 92.69 (±0.14) |
| 1B | MLM | 20% | 93.03 (±0.15) | 92.28 (±0.30) | 92.84 (±0.31) | 92.72 (±0.19) |
| | | 30% | 92.93 (±0.17) | 92.06 (±0.10) | 92.55 (±0.16) | 92.51 (±0.09) |
| | | 40% | 92.98 (±0.22) | 92.05 (±0.14) | 92.51 (±0.47) | 92.51 (±0.16) |
| | | 50% | **93.12** (±0.25) | **92.46** (±0.30) | **93.06** (±0.55) | **92.88** (±0.19) |
| | CLM | - | 92.60 (±0.09) | 91.93 (±0.19) | 92.84 (±0.12) | 92.46 (±0.10) |

Table 3: MLM vs. CLM downstream performance results on TC datasets.

## C.3 CLM-TO-MLM RATIO

This appendix provides detailed results in the PFS setting, analyzing the impact of varying the CLM-MLM step ratio during pretraining. Results are presented for 610M models with 40% masking for MLM. Table 13, Table 11, Table 12, and Table 13 present dataset-level results for SC, TC, QA, and IR, respectively.

## C.4 CONTINUED PRETRAINING

This appendix presents detailed findings for the CPT setting, evaluating different CPT lengths. All experiments use 610M-parameter models, with MLM reported for a 40% masking ratio. Table 14, Table 15, Table 16, and Table 17 summarize dataset-level performance for SC, TC, QA, and IR, respectively.

| Model Size | Objective | Masking | SQuAD | SQuAD-v2 | ReCoRD | Average |
|---|---|---|---|---|---|---|
| 210M | MLM | 20% | **74.66** (±0.26) | **64.39** (±0.44) | 16.15 (±6.01) | **51.73** (±2.03) |
| | | 30% | 74.59 (±0.11) | 61.87 (±1.02) | **16.57** (±7.13) | 51.01 (±2.58) |
| | | 40% | 73.73 (±0.11) | 61.10 (±1.34) | 13.45 (±2.93) | 49.43 (±1.36) |
| | | 50% | 72.75 (±0.50) | 58.47 (±1.96) | 14.74 (±6.65) | 48.65 (±2.99) |
| | CLM | - | 64.00 (±0.54) | 52.60 (±0.69) | 1.17 (±0.78) | 39.26 (±0.32) |
| 610M | MLM | 20% | 76.97 (±0.19) | **71.31** (±0.30) | **46.74** (±1.11) | **65.00** (±0.31) |
| | | 30% | 77.51 (±0.20) | 65.98 (±2.56) | 40.92 (±1.75) | 61.47 (±0.61) |
| | | 40% | **77.76** (±0.29) | 70.63 (±0.58) | 39.93 (±5.26) | 62.77 (±1.86) |
| | | 50% | 77.41 (±0.47) | 68.61 (±0.53) | 45.82 (±0.63) | 63.95 (±0.47) |
| | CLM | - | 69.42 (±0.30) | 56.44 (±0.10) | 0.42 (±0.39) | 42.09 (±0.20) |
| 1B | MLM | 20% | 77.81 (±0.57) | 73.03 (±0.35) | 54.38 (±0.92) | 68.41 (±0.33) |
| | | 30% | 76.82 (±0.61) | 66.09 (±2.13) | 43.13 (±4.94) | 62.02 (±1.84) |
| | | 40% | **79.93** (±0.27) | **74.96** (±0.41) | **55.95** (±0.99) | **70.28** (±0.32) |
| | | 50% | 79.62 (±0.27) | 73.53 (±0.46) | 55.67 (±1.19) | 69.61 (±0.56) |
| | CLM | - | 70.54 (±0.69) | 57.99 (±0.96) | 4.90 (±4.31) | 44.48 (±1.65) |

Table 4: MLM vs. CLM downstream performance results on QA datasets.

| Model Size | Objective | Masking | NQ | MS MARCO | MLDR | Average |
|---|---|---|---|---|---|---|
| 210M | MLM | 20% | 80.23 (±0.54) | 90.24 (±0.76) | 53.76 (±0.90) | 74.74 (±0.35) |
| | | 30% | 81.20 (±0.92) | **90.44** (±0.75) | 55.14 (±1.02) | 75.59 (±0.57) |
| | | 40% | 81.43 (±1.35) | 89.35 (±0.98) | 56.56 (±0.62) | 75.78 (±0.43) |
| | | 50% | **82.97** (±0.42) | 89.31 (±0.35) | **58.85** (±0.46) | **77.04** (±0.27) |
| | CLM | - | 82.08 (±0.93) | 86.45 (±0.38) | 49.95 (±1.86) | 72.83 (±0.86) |
| 610M | MLM | 20% | 85.14 (±0.49) | 88.82 (±0.78) | 57.07 (±0.93) | 77.01 (±0.52) |
| | | 30% | 83.47 (±0.93) | 90.36 (±0.81) | 58.65 (±0.42) | 77.50 (±0.42) |
| | | 40% | 86.76 (±0.63) | **91.65** (±0.45) | 60.25 (±0.76) | 79.55 (±0.35) |
| | | 50% | **87.64** (±0.74) | 90.73 (±1.26) | **60.49** (±0.73) | **79.62** (±0.54) |
| | CLM | - | 85.57 (±0.34) | 89.88 (±1.17) | 53.16 (±0.42) | 76.20 (±0.54) |
| 1B | MLM | 20% | 86.02 (±0.55) | 91.78 (±0.56) | 59.85 (±0.40) | 79.22 (±0.19) |
| | | 30% | 86.36 (±0.54) | 90.90 (±0.61) | 61.11 (±0.25) | 79.46 (±0.04) |
| | | 40% | 87.89 (±0.66) | 91.45 (±0.40) | 60.51 (±0.80) | 79.95 (±0.41) |
| | | 50% | **89.28** (±0.40) | **92.98** (±0.65) | **63.64** (±0.34) | **81.97** (±0.27) |
| | CLM | - | 88.73 (±0.54) | 90.85 (±1.22) | 56.45 (±0.39) | 78.68 (±0.55) |

Table 5: MLM vs. CLM downstream performance results on IR datasets.

# D  ADDITIONAL RESULTS

This appendix provides additional results that complement our main analysis. Figure 10 complements Figure 4 by showing that CLM is not only more data-efficient than MLM but also more efficient in terms of training FLOPs. Figure 11 confirms, at the 1B scale, the same trends observed for the 610M model in Figure 6. Figure 12 presents loss curves across different CPT training budgets, starting from either an MLM- or CLM-pre-trained model and then continuing with MLM.

| Objective | Training Step | SST-2 | QQP | MNLI | Average |
|---|---|---|---|---|---|
| MLM | 1K | 81.06 (±0.63) | 73.78 (±0.18) | 45.90 (±0.22) | 66.91 (±0.26) |
| | 2K | 81.40 (±0.81) | 73.88 (±0.12) | 49.40 (±0.33) | 68.23 (±0.34) |
| | 5K | 82.11 (±0.62) | 78.44 (±1.06) | 56.90 (±2.44) | 72.49 (±1.17) |
| | 10K | 86.90 (±1.93) | 83.83 (±0.27) | 73.37 (±1.62) | 81.37 (±0.90) |
| | 20K | 91.38 (±0.54) | 85.97 (±0.28) | 79.36 (±0.17) | 85.57 (±0.20) |
| | 40K | 90.94 (±0.67) | 86.42 (±0.25) | 81.84 (±0.30) | 86.40 (±0.27) |
| CLM | 1K | 79.52 (±1.00) | 73.56 (±0.24) | 51.89 (±0.49) | 68.32 (±0.28) |
| | 2K | 84.24 (±0.47) | 76.03 (±0.70) | 60.54 (±0.89) | 73.61 (±0.26) |
| | 5K | 87.11 (±0.93) | 81.80 (±0.31) | 66.76 (±0.34) | 78.56 (±0.46) |
| | 10K | 89.33 (±0.29) | 82.66 (±0.24) | 69.75 (±0.22) | 80.58 (±0.14) |
| | 20K | 90.50 (±0.35) | 83.74 (±0.08) | 73.87 (±0.45) | 82.70 (±0.07) |
| | 40K | 90.94 (±0.43) | 84.20 (±0.21) | 75.31 (±0.43) | 83.48 (±0.21) |

Table 6: MLM vs. CLM downstream performance on SC datasets at various pretraining checkpoints. Results correspond to 610M models with MLM using 40% masking.

| Objective | Training Step | OntoNotes | CoNLL | UNER | Average |
|---|---|---|---|---|---|
| MLM | 1K | 84.39 (±0.45) | 82.47 (±0.28) | 82.58 (±0.78) | 83.15 (±0.26) |
| | 2K | 88.64 (±0.16) | 86.02 (±0.19) | 85.74 (±0.64) | 86.80 (±0.23) |
| | 5K | 90.93 (±0.07) | 88.10 (±0.94) | 89.05 (±0.89) | 89.36 (±0.61) |
| | 10K | 92.03 (±0.14) | 90.60 (±0.22) | 91.31 (±0.13) | 91.31 (±0.08) |
| | 20K | 92.62 (±0.07) | 91.36 (±0.21) | 92.06 (±0.12) | 92.02 (±0.03) |
| | 40K | 92.86 (±0.17) | 91.45 (±0.28) | 91.75 (±0.59) | 92.02 (±0.19) |
| CLM | 1K | 88.09 (±0.13) | 85.51 (±0.19) | 87.44 (±0.35) | 87.01 (±0.15) |
| | 2K | 90.16 (±0.11) | 88.82 (±0.21) | 89.57 (±0.31) | 89.52 (±0.15) |
| | 5K | 91.54 (±0.10) | 90.82 (±0.29) | 91.15 (±0.28) | 91.17 (±0.08) |
| | 10K | 92.27 (±0.11) | 91.47 (±0.18) | 92.49 (±0.22) | 92.08 (±0.10) |
| | 20K | 92.69 (±0.08) | 92.21 (±0.16) | 92.99 (±0.30) | 92.63 (±0.12) |
| | 40K | 92.75 (±0.11) | 92.25 (±0.14) | 92.97 (±0.06) | 92.65 (±0.03) |

Table 7: MLM vs. CLM downstream performance on TC datasets at various pretraining checkpoints. Results correspond to 610M models with MLM using 40% masking.

| Objective | Training Step | SQuAD | SQuAD-v2 | ReCoRD | Average |
|---|---|---|---|---|---|
| MLM | 1K | 0.00 (±0.00) | 50.07 (±0.00) | 0.00 (±0.00) | 16.69 (±0.00) |
| | 2K | 0.94 (±0.50) | 50.07 (±0.00) | 0.00 (±0.00) | 17.00 (±0.17) |
| | 5K | 33.80 (±15.60) | 49.87 (±0.27) | 0.01 (±0.01) | 27.89 (±5.17) |
| | 10K | 59.19 (±12.12) | 51.43 (±1.79) | 4.38 (±2.42) | 38.33 (±4.13) |
| | 20K | 73.09 (±0.29) | 58.38 (±0.63) | 13.79 (±3.53) | 48.42 (±1.27) |
| | 40K | 77.00 (±0.27) | 68.72 (±0.87) | 27.98 (±10.00) | 57.90 (±3.27) |
| CLM | 1K | 4.52 (±0.75) | 50.00 (±0.09) | 0.00 (±0.00) | 18.17 (±0.24) |
| | 2K | 30.93 (±11.46) | 49.65 (±0.27) | 0.10 (±0.05) | 26.89 (±3.86) |
| | 5K | 52.27 (±0.76) | 50.52 (±0.49) | 0.11 (±0.11) | 34.30 (±0.11) |
| | 10K | 62.31 (±0.51) | 53.17 (±0.52) | 0.32 (±0.07) | 38.60 (±0.32) |
| | 20K | 66.68 (±0.57) | 54.99 (±0.62) | 0.57 (±0.29) | 40.75 (±0.19) |
| | 40K | 69.40 (±0.38) | 56.17 (±0.47) | 0.41 (±0.08) | 41.99 (±0.13) |

Table 8: MLM vs. CLM downstream performance on QA datasets at various pretraining checkpoints. Results correspond to 610M models with MLM using 40% masking.

| Objective | Training Step | NQ | MS MARCO | MLDR | Average |
|---|---|---|---|---|---|
| MLM | 1K | 8.41 (±0.61) | 43.97 (±2.88) | 8.20 (±0.87) | 20.19 (±1.38) |
|  | 2K | 21.64 (±0.82) | 54.88 (±1.33) | 15.15 (±0.97) | 30.55 (±0.84) |
|  | 5K | 50.47 (±0.74) | 78.14 (±0.59) | 34.89 (±0.69) | 54.50 (±0.44) |
|  | 10K | 69.32 (±0.90) | 85.35 (±1.73) | 44.30 (±0.57) | 66.33 (±0.61) |
|  | 20K | 82.50 (±0.54) | 89.60 (±0.65) | 56.36 (±1.19) | 76.16 (±0.62) |
|  | 40K | 85.46 (±0.24) | 91.50 (±0.54) | 60.83 (±0.29) | 79.26 (±0.23) |
| CLM | 1K | 22.24 (±0.92) | 54.20 (±1.70) | 13.39 (±0.79) | 29.94 (±0.49) |
|  | 2K | 45.12 (±0.68) | 72.93 (±3.32) | 25.26 (±0.87) | 47.77 (±1.09) |
|  | 5K | 70.67 (±0.89) | 85.19 (±0.81) | 39.79 (±1.47) | 65.22 (±0.60) |
|  | 10K | 78.47 (±1.94) | 88.23 (±0.68) | 46.14 (±1.36) | 70.95 (±1.05) |
|  | 20K | 83.80 (±0.88) | 89.49 (±0.75) | 53.38 (±0.68) | 75.56 (±0.45) |
|  | 40K | 84.88 (±0.71) | 90.52 (±0.83) | 52.76 (±1.65) | 76.05 (±0.59) |

Table 9: MLM vs. CLM downstream performance on IR datasets at various pretraining checkpoints. Results correspond to 610M models with MLM using 40% masking.

| Total Steps | CLM+MLM Mix | SST-2 | QQP | MNLI | Average |
|---|---|---|---|---|---|
| 12K | 0K+12K | 89.33 (±0.68) | 84.62 (±0.18) | 75.17 (±0.33) | 83.04 (±0.28) |
|  | 3K+9K | **90.09** (±0.45) | **84.59** (±0.10) | **76.32** (±0.36) | **83.67** (±0.19) |
|  | 6K+6K | 90.02 (±0.34) | 84.70 (±0.16) | 76.15 (±0.32) | 83.63 (±0.17) |
|  | 9K+3K | 89.77 (±0.77) | 83.93 (±0.07) | 73.25 (±0.15) | 82.32 (±0.25) |
|  | 12K+0K | 89.66 (±0.51) | 82.54 (±0.13) | 70.40 (±0.38) | 80.86 (±0.13) |
| 22K | 0K+22K | 91.63 (±0.30) | 86.30 (±0.08) | 80.27 (±0.21) | 86.07 (±0.07) |
|  | 5K+17K | 91.28 (±0.80) | 86.42 (±0.32) | **81.24** (±0.31) | **86.31** (±0.29) |
|  | 11K+11K | **91.90** (±0.71) | **86.69** (±0.12) | 80.39 (±0.22) | 86.33 (±0.25) |
|  | 17K+5K | 90.94 (±0.40) | 85.86 (±0.16) | 79.07 (±0.07) | 85.29 (±0.18) |
|  | 22K+0K | 90.80 (±0.22) | 83.86 (±0.17) | 74.26 (±0.27) | 82.97 (±0.15) |
| 42K | 0K+42K | 91.42 (±0.62) | 86.98 (±0.15) | 82.59 (±0.16) | 87.00 (±0.18) |
|  | 10K+32K | **92.41** (±0.18) | **87.32** (±0.23) | **83.83** (±0.25) | **87.85** (±0.13) |
|  | 21K+21K | 92.36 (±0.39) | 87.11 (±0.18) | 83.28 (±0.14) | 87.58 (±0.11) |
|  | 32K+10K | 92.16 (±0.43) | 87.14 (±0.10) | 81.90 (±0.24) | 87.06 (±0.15) |
|  | 42K+0K | 91.10 (±0.39) | 84.14 (±0.12) | 75.49 (±0.22) | 83.58 (±0.17) |

Table 10: Impact of the CLM-to-MLM pretraining steps ratio on downstream performance in the PFS setup for SC datasets. Results are reported for 610M models with MLM using 40% masking.

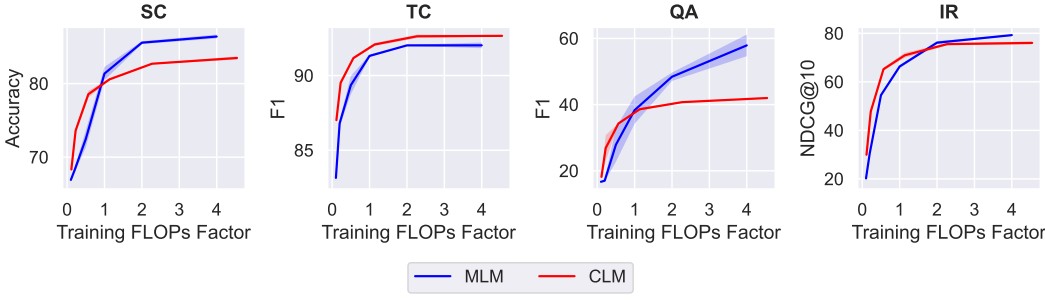

Figure 10: Downstream performance with respect to pretraining FLOPs for CLM and MLM. Results are reported for 610M models, with a 40% masking ratio for MLM. Training FLOPs are computed following the formula provided by Hoffmann et al. (2022).

| Total Steps | CLM+MLM Mix | OntoNotes | CoNLL | UNER | Average |
|---|---|---|---|---|---|
| | 0K+12K | 92.32 (±0.15) | 90.81 (±0.22) | 91.59 (±0.20) | 91.58 (±0.10) |
| | 3K+9K | 92.64 (±0.10) | 91.63 (±0.09) | 92.20 (±0.24) | 92.16 (±0.11) |
| 12K | 6K+6K | 92.69 (±0.12) | 91.62 (±0.20) | 92.15 (±0.23) | 92.15 (±0.09) |
| | 9K+3K | **92.79** (±0.06) | **91.83** (±0.26) | **92.96** (±0.15) | **92.53** (±0.10) |
| | 12K+0K | 92.29 (±0.09) | 91.67 (±0.25) | 92.47 (±0.22) | 92.14 (±0.06) |
| | 0K+22K | 92.83 (±0.15) | 91.62 (±0.14) | 92.39 (±0.24) | 92.28 (±0.15) |
| | 5K+17K | 93.09 (±0.08) | 92.15 (±0.16) | **93.08** (±0.53) | 92.77 (±0.13) |
| 22K | 11K+11K | **93.15** (±0.04) | **92.42** (±0.25) | 92.93 (±0.30) | **92.83** (±0.08) |
| | 17K+5K | 93.13 (±0.12) | 92.46 (±0.18) | 92.86 (±0.24) | 92.82 (±0.12) |
| | 22K+0K | 92.73 (±0.06) | 92.22 (±0.37) | 92.94 (±0.52) | 92.63 (±0.15) |
| | 0K+42K | 92.97 (±0.15) | 91.58 (±0.44) | 92.07 (±0.77) | 92.21 (±0.26) |
| | 10K+32K | **93.35** (±0.07) | 92.39 (±0.20) | **93.45** (±0.36) | **93.07** (±0.15) |
| 42K | 21K+21K | 93.21 (±0.15) | **92.43** (±0.13) | 93.02 (±0.43) | 92.89 (±0.15) |
| | 32K+10K | 93.32 (±0.04) | 92.41 (±0.17) | 93.02 (±0.24) | 92.92 (±0.11) |
| | 42K+0K | 92.79 (±0.05) | 92.16 (±0.21) | 93.12 (±0.29) | 92.69 (±0.14) |

Table 11: Impact of the CLM-to-MLM pretraining steps ratio on downstream performance in the PFS setup for TC datasets. Results are reported for 610M models with MLM using 40% masking.

| Total Steps | CLM+MLM Mix | SQuAD | SQuAD-v2 | ReCoRD | Average |
|---|---|---|---|---|---|
| | 0K+12K | 67.95 (±0.65) | 53.73 (±2.04) | 2.85 (±1.06) | 41.51 (±0.55) |
| | 3K+9K | **71.36** (±0.63) | **56.28** (±0.44) | **3.68** (±1.81) | **43.77** (±0.63) |
| 12K | 6K+6K | 71.29 (±0.27) | 55.85 (±0.89) | 1.19 (±0.40) | 42.78 (±0.37) |
| | 9K+3K | 68.20 (±0.83) | 53.85 (±1.09) | 2.55 (±4.07) | 41.53 (±1.81) |
| | 12K+0K | 63.40 (±0.72) | 53.49 (±0.40) | 0.32 (±0.11) | 39.07 (±0.34) |
| | 0K+22K | 74.02 (±0.38) | 61.02 (±0.92) | 19.29 (±5.45) | 51.44 (±1.81) |
| | 5K+17K | **75.71** (±0.93) | 61.50 (±2.71) | 26.08 (±6.25) | 54.43 (±1.51) |
| 22K | 11K+11K | 75.43 (±0.21) | **61.33** (±0.73) | **31.14** (±6.92) | **55.97** (±2.43) |
| | 17K+5K | 74.15 (±0.46) | 59.03 (±1.53) | 26.99 (±3.43) | 53.39 (±1.31) |
| | 22K+0K | 67.52 (±0.40) | 55.44 (±0.66) | 0.47 (±0.14) | 41.14 (±0.25) |
| | 0K+42K | 77.76 (±0.29) | 70.63 (±0.58) | 39.93 (±5.26) | 62.77 (±1.86) |
| | 10K+32K | **79.20** (±0.22) | **72.33** (±0.81) | **51.79** (±1.50) | **67.77** (±0.53) |
| 42K | 21K+21K | 77.12 (±0.28) | 67.81 (±0.66) | 33.60 (±11.35) | 59.51 (±3.84) |
| | 32K+10K | 76.54 (±0.31) | 64.49 (±0.82) | 43.20 (±1.29) | 61.41 (±0.66) |
| | 42K+0K | 69.42 (±0.30) | 56.44 (±0.10) | 0.42 (±0.39) | 42.09 (±0.20) |

Table 12: Impact of the CLM-to-MLM pretraining steps ratio on downstream performance in the PFS setup for QA datasets. Results are reported for 610M models with MLM using 40% masking.

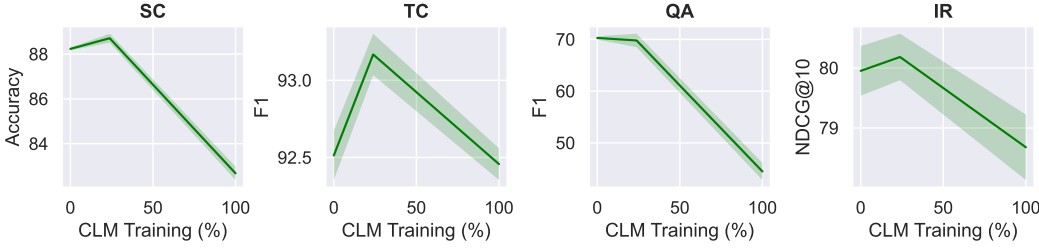

Figure 11: Impact of two-stage CLM+MLM pretraining on downstream performance after 42,000 training steps. Results are reported for the 1B model, with MLM applied using a 40% masking ratio. To keep computation costs manageable, CLM+MLM results are shown only for the 25%-75% regime.

| Total Steps | CLM+MLM Mix | NQ | MS MARCO | MLDR | Average |
|---|---|---|---|---|---|
| 12K | 0K+12K | 74.15 (±1.46) | 85.95 (±1.83) | 49.59 (±0.56) | 69.90 (±1.21) |
| | 3K+9K | 80.08 (±0.44) | **89.63** (±0.87) | 55.09 (±0.56) | 74.94 (±0.22) |
| | 6K+6K | **80.67** (±0.61) | 88.30 (±0.79) | **56.70** (±0.58) | **75.22** (±0.41) |
| | 9K+3K | 80.10 (±0.44) | 89.23 (±1.28) | 55.63 (±0.87) | 74.99 (±0.31) |
| | 12K+0K | 80.16 (±1.16) | 88.88 (±1.27) | 49.26 (±0.54) | 72.77 (±0.82) |
| 22K | 0K+22K | 83.72 (±0.85) | 90.42 (±0.38) | 58.47 (±0.45) | 77.54 (±0.28) |
| | 5K+17K | 84.07 (±0.52) | 89.38 (±0.77) | 58.42 (±1.00) | 77.29 (±0.46) |
| | 11K+11K | **84.48** (±0.50) | 89.72 (±0.56) | 58.22 (±0.47) | 77.47 (±0.40) |
| | 17K+5K | 83.36 (±0.57) | **90.97** (±0.56) | **59.12** (±0.26) | **77.81** (±0.18) |
| | 22K+0K | 84.01 (±0.75) | 90.36 (±0.98) | 52.75 (±2.03) | 75.71 (±0.34) |
| 42K | 0K+42K | 86.76 (±0.63) | 91.65 (±0.45) | 60.25 (±0.76) | 79.55 (±0.35) |
| | 10K+32K | 86.79 (±0.28) | **91.98** (±1.35) | 60.86 (±1.77) | **79.88** (±0.87) |
| | 21K+21K | 86.80 (±0.86) | 90.98 (±0.92) | **61.22** (±1.27) | 79.67 (±0.81) |
| | 32K+10K | **86.82** (±0.49) | 91.40 (±0.92) | 61.14 (±0.84) | 79.79 (±0.36) |
| | 42K+0K | 85.57 (±0.34) | 89.88 (±1.17) | 53.16 (±0.42) | 76.20 (±0.54) |

Table 13: Impact of the CLM-to-MLM pretraining steps ratio on downstream performance in the PFS setup for IR datasets. Results are reported for 610M models with MLM using 40% masking.

| PT Objective | CPT Steps | SST-2 | QQP | MNLI | Average |
|---|---|---|---|---|---|
| MLM | - | 91.42 (±0.62) | 86.98 (±0.15) | 82.59 (±0.16) | 87.00 (±0.18) |
| | 2K | 91.54 (±0.54) | 86.84 (±0.16) | 82.85 (±0.27) | 87.08 (±0.20) |
| | 12K | 92.71 (±0.37) | 86.89 (±0.31) | 83.23 (±0.29) | 87.61 (±0.21) |
| | 22K | 92.16 (±0.59) | 86.94 (±0.10) | 83.31 (±0.35) | 87.47 (±0.24) |
| CLM | - | 91.10 (±0.39) | 84.14 (±0.12) | 75.49 (±0.22) | 83.58 (±0.17) |
| | 2K | 92.06 (±0.48) | 85.93 (±0.15) | 78.18 (±0.23) | 85.39 (±0.17) |
| | 12K | 92.98 (±0.34) | 87.48 (±0.13) | 83.05 (±0.18) | 87.84 (±0.15) |
| | 22K | **93.62** (±0.33) | **87.56** (±0.08) | **84.02** (±0.10) | **88.40** (±0.11) |

Table 14: Impact of continued MLM pretraining on both MLM- and CLM-pretrained models across different sequence lengths on SC datasets. Results are reported for 610M models with 40% masking for MLM.

| PT Objective | CPT Steps | OntoNotes | CoNLL | UNER | Average |
|---|---|---|---|---|---|
| MLM | - | **92.97** (±0.15) | 91.58 (±0.44) | 92.07 (±0.77) | 92.21 (±0.26) |
| | 2K | 92.85 (±0.26) | 90.89 (±1.10) | 92.74 (±0.39) | 92.16 (±0.41) |
| | 12K | 93.02 (±0.17) | 91.19 (±0.82) | 92.59 (±0.51) | 92.27 (±0.41) |
| | 22K | 92.82 (±0.45) | 91.41 (±0.16) | 92.10 (±0.57) | 92.11 (±0.32) |
| CLM | - | 92.79 (±0.05) | 92.16 (±0.21) | **93.12** (±0.29) | 92.69 (±0.14) |
| | 2K | 92.91 (±0.07) | **92.71** (±0.26) | 93.02 (±0.26) | **92.88** (±0.13) |
| | 12K | 92.78 (±0.56) | 92.53 (±0.13) | 93.01 (±0.29) | 92.77 (±0.12) |
| | 22K | 92.85 (±0.13) | 92.30 (±0.19) | 92.94 (±0.11) | 92.70 (±0.05) |

Table 15: Impact of continued MLM pretraining on both MLM- and CLM-pretrained models across different sequence lengths on TC datasets. Results are reported for 610M models with 40% masking for MLM.

| PT Objective | CPT Steps | SQuAD | SQuAD-v2 | ReCoRD | Average |
|---|---|---|---|---|---|
| MLM | - | 77.76 (±0.29) | 70.63 (±0.58) | 39.93 (±5.26) | 62.77 (±1.86) |
| | 2K | 77.92 (±0.21) | 71.07 (±0.85) | 41.69 (±4.59) | 63.56 (±1.77) |
| | 12K | 78.71 (±0.23) | 72.35 (±0.67) | 48.66 (±2.17) | 66.57 (±0.85) |
| | 22K | **79.01** (±0.18) | **72.77** (±0.47) | 51.21 (±1.32) | **67.66** (±0.53) |
| CLM | - | 69.42 (±0.30) | 56.44 (±0.10) | 0.42 (±0.39) | 42.09 (±0.20) |
| | 2K | 73.48 (±0.28) | 59.13 (±0.21) | 10.94 (±5.90) | 47.85 (±2.04) |
| | 12K | 77.25 (±0.29) | 65.62 (±1.15) | 46.75 (±1.38) | 63.21 (±0.17) |
| | 22K | 78.13 (±0.27) | 69.69 (±0.56) | **52.06** (±0.72) | 66.62 (±0.34) |

Table 16: Impact of continued MLM pretraining on both MLM- and CLM-pretrained models across different sequence lengths on QA datasets. Results are reported for 610M models with 40% masking for MLM.

| PT Objective | CPT Steps | NQ | MS MARCO | MLDR | Average |
|---|---|---|---|---|---|
| MLM | - | 86.76 (±0.63) | 91.65 (±0.45) | 60.25 (±0.76) | 79.55 (±0.35) |
| | 2K | 86.77 (±0.84) | 92.18 (±0.63) | 62.31 (±0.42) | 80.42 (±0.27) |
| | 12K | 87.01 (±0.52) | 92.32 (±0.92) | 60.44 (±1.02) | 79.92 (±0.47) |
| | 22K | 87.58 (±0.62) | **92.52** (±0.79) | 61.26 (±0.53) | 80.45 (±0.27) |
| CLM | - | 85.57 (±0.34) | 89.88 (±1.17) | 53.16 (±0.42) | 76.20 (±0.54) |
| | 2K | 86.46 (±0.38) | 88.80 (±0.76) | 60.45 (±0.43) | 78.57 (±0.22) |
| | 12K | **88.29** (±0.26) | 89.56 (±0.92) | 62.97 (±0.52) | 80.27 (±0.29) |
| | 22K | 87.67 (±0.93) | 90.71 (±0.23) | **63.72** (±0.41) | **80.70** (±0.26) |

Table 17: Impact of continued MLM pretraining on both MLM- and CLM-pretrained models across different sequence lengths on IR datasets. Results are reported for 610M models with 40% masking for MLM.

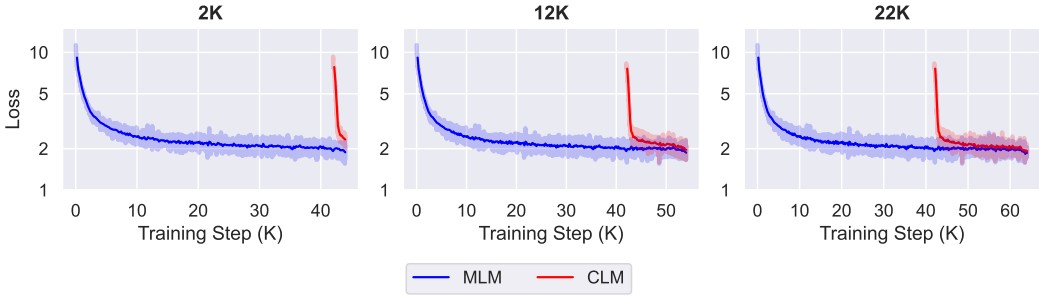

Figure 12: Loss trends in the CPT setting. The blue curve shows the loss trajectory of a model using MLM for both pretraining and continued pretraining, while the red curve depicts a CLM model adapted using the MLM objective during CPT. Results are reported for the 610M model with a 40% masking ratio for MLM.

