# OpenReview forum: "Should We Still Pretrain Encoders with Masked Language Modeling?"
_ICLR.cc/2026/Conference — ICLR 2026 Poster_

### Official Review · Reviewer_hZNQ · 2025-10-31

**Soundness:** 3
**Presentation:** 3
**Contribution:** 3
**Rating:** 6
**Confidence:** 3

**Summary:**

This paper conducts a comprehensive empirical comparison between Masked Language Modeling (MLM) and Causal Language Modeling (CLM) for encoder pretraining. Using 38 models (210M–1B parameters) trained under matched computational budgets, the authors analyze performance across multiple NLP tasks and setups, including two-stage (CLM→MLM) and continued pretraining. The results show that MLM remains generally superior for text representation tasks, while CLM exhibits higher data efficiency and fine-tuning stability. Combining both (CLM followed by MLM) achieves the best overall trade-off between efficiency and final performance.

**Strengths:**

1. The paper is clearly written and logically organized. The narrative is easy to follow, and figures/tables effectively support the main findings.
2. The experimental analysis is extensive and carefully controlled. The authors evaluate across diverse model sizes, masking ratios, and task types, ensuring that performance differences stem from the pretraining objective itself rather than confounding factors.
3. The research question is highly relevant: as decoder-based models increasingly dominate embedding benchmarks, understanding whether CLM pretraining offers intrinsic advantages is crucial. The paper addresses this gap convincingly and provides practical insights for future encoder design.

**Weaknesses:**

1. The paper establishes that MLM excels in tasks requiring bidirectional reasoning (e.g., QA, classification), while CLM performs competitively in token-level or retrieval settings. However, the authors stop short of elaborating on why certain pretraining paradigms align better with specific task types. A deeper discussion of this task–objective relationship would make the work more insightful and actionable for practitioners.
2. Although training budgets are said to be matched, the paper does not provide explicit comparisons in FLOPs, runtime, or memory efficiency. Such quantitative reporting would strengthen the claims about CLM’s data and compute efficiency.

**Questions:**

please see weakness.

---

> ### Author Response · Authors · 2025-11-26
>
> We thank the reviewer for their feedback and appreciate that they found the clarity, relevance, and experimental setup of our work compelling. We also thank them for their thoughtful remarks, which we address below.
>
> * **"A deeper discussion of this task-objective relationship would make the work more insightful and actionable for practitioners."**
>
> When comparing plain CLM and plain MLM in Section 3, we observed that for TC tasks, CLM performed on par with MLM. Intuitively, this may be explained by the nature of token-level tasks, which rely less on bidirectionally informed context for identifying named entities or word types. For example, a named entity can often be recognized in isolation and does not require extensive contextualization to be correctly identified.
>
> While exploring the underlying reasons for this observation is an interesting direction for future work, ***which we now highlight explicitly in Section 8***, we consider it beyond the scope of our study. Nonetheless, we think our findings still provide actionable insights for practitioners. For industry-level tasks such as named entity recognition, this suggests that the most effective backbone is not necessarily a pre-trained encoder, and that decoders can also be strong alternatives. Furthermore, as shown in Section 4, sequentially combining CLM and MLM consistently outperforms MLM-only pretraining, offering a ready-to-use strategy for practitioners aiming to pretrain encoder models from scratch.
>
> * **"The paper does not provide explicit comparisons in FLOPs, runtime, or memory efficiency."**
>
> In our study, we consider data access to be the primary limiting factor and therefore report efficiency in terms of data sample efficiency. Nonetheless, we agree that complementing Figure 4 with an analysis based on training FLOPs provides valuable additional context.
>
> Using the FLOPs estimation formula from [1], we find that a training step with CLM is approximately 1.13x more compute-intensive than with MLM at a 40% masking ratio. This results in a slight rightward offset of the CLM curve (***see Figure 10 in Appendix D, which we added in the updated paper version***), reflecting that CLM involves backpropagation over all tokens rather than only the masked subset.
>
> However, this difference does not affect the main conclusion: CLM remains more efficient in the early stages of training, both in terms of data sample usage and total training FLOPs.
>
> We thank the reviewer again for their input and believe our answers address their concerns. We remain at their disposal for any additional clarification.
>
> **References:**
>
> [1] Training Compute-Optimal Large Language Models (Hoffman et al., 2022)

---

### Official Review · Reviewer_e3tm · 2025-10-31

**Soundness:** 3
**Presentation:** 3
**Contribution:** 3
**Rating:** 6
**Confidence:** 4

**Summary:**

This paper compares MLM (masked language modeling) and CLM (causal language model) objectives in a large variety of scenarios. As the authors note, encoder models are typically trained solely with the MLM objective to encourage good representations, while decoder models use CLM towards the direct objective of next token predicting. The authors design an experimentation setup to do fair assessments of both of these methods, carefully ablating aspects like model size, masking rate, and curriculum incorporating each aspect. They test on a variety of common evaluations to find cases where different models have an advantage. This leads to some surprising conclusions around training encoder models in a data and compute efficient manner - specifically, that a "biphasic training strategy" can lead to good performance at the scales tested here. I also found the analysis of fine-tuning stability particularly valuable, and somewhat under-considered in other works.

**Strengths:**

This paper has many interesting experiments and appears to explore a large space of possible model configurations.

- Reasonable experimental setup using common best practices in small model encoder/decoder training: high quality data choice, learning rate schedule, controlled architecture

- Carefully sweeps an interesting set of parameters: model size, masking ratio, curricula combining CLM + MLM at different ratios, careful task specific tuning to analyze model stability. It is uncommon to see all of these varied at reasonable experimental scale, so the authors should be commended.

- The decision to test mixed objectives like applying MLM after CLM in the same training run is very interesting. The authors note that the this method is applicable to the case where an encoder model can be trained from an existing pretrained decoder only model, as it seems many industrial resources are focused on decoder training.

- The experiments and finding seem very practically motivated. Finetuning stability is an important practical aspect of encoder models as commonly used, and a focus on limited data efficiency is also practical.

**Weaknesses:**

My main weaknesses with this paper involve data choices and budget. This paper has many interesting experiments, some of which lead to further research questions.

1. As the authors acknowledge in section 8, their dataset is limited to 100B tokens. While this is far beyond the decoder compute optimal ratio, it does not seem well established whether this is a reasonable budget for encoder model training. For example, the progression from the original BERT to RoBERTA (and XLM-R) involved a large increase in data scale. Further recent research has shown that considering inference compute costs pushes scaling to favor longer data durations with smaller models ("overtraining" - see the llama3 paper or https://arxiv.org/abs/2401.00448). There is closely related work (only briefly cited by the paper authors, possibly concurrent) that trains similar model sizes up to 2T tokens: https://arxiv.org/abs/2507.11412 (Weller et al. 2025).

2. Related, the choice of 100B tokens is only focused on the fineweb edu data. This data is high quality for some domains, but does not represent the diversity of data present in a practical encoder data mixture. See the previously mentioned work or the ModernBERT paper (https://arxiv.org/abs/2412.13663), which additionally includes code and scientific papers. The original Fineweb edu work shows the greatest improvements in benchmarks like MMLU, which specifically draw from education-relevant sources.

3. The total size of 100B tokens also limits the experimental ratios when combining MLM + CLM objectives. While motivated by using an existing pretrained LLM, the ratios of 25% budget increments (corresponding to 25B tokens), seem unrealistic. This would mean that the 75-25% scenario uses CLM on 75B tokens and MLM on 25B, which seems far from the more realistic scenario of a very large budget allocated to CLM with a smaller scale MLM.

In a similar vein, while the masking ratio experiments are very interesting, it is hard to judge whether the results are due to the relatively short training budgets. Perhaps 100B tokens is not enough to learn good representations and the very high or low rates. As the authors note in line 473:

> but learning dynamics suggest that further performance improvements are possible with extended training

Minor:

Further analysis or explanations of differences in IR and token level tasks (253-254). Other analysis of the interaction between LR decay and curriculum would be interesting

The tables in the appendix are appreciated but would benefit from formatting such as bolding extreme values.

**Questions:**

- Any further analysis or intuition on the u-shaped masking ratio?

---

> ### Author Response · Authors · 2025-11-26
>
> We sincerely thank the reviewer for their comprehensive feedback and thoughtful remarks, which we address below. We are pleased that they appreciate the extensiveness of our experiments and highlight the practical value of our findings, particularly regarding data efficiency and fine-tuning stability.
>
> **Major comments:**
>
> * **"Their dataset is limited to 100B tokens."**
>
> The reviewer’s point is indeed valid, when the objective is to train a model achieving state of the art performance. However, our goal was not to build a state of the art model. Rather, we aimed to create an experimental setup with a pretraining duration long enough to yield interpretable results across many training configurations, enabling us to extract practical and useful insights for practitioners.
>
> As noted, and as the reviewer points out, training on 100B tokens represents roughly five times the compute-optimal budget for decoders in the 1B-parameter range [1]. While this may be insufficient to achieve state-of-the-art performance [2], we believe the setup still provides meaningful signal [3] while remaining computationally tractable. In our work, scaling (both in terms of model size and training compute) does not appear to substantially alter our observations.
>
> * **"There is closely related work that trains similar model sizes up to 2T tokens."**
>
> We thank the reviewer for highlighting this line of work, which is highly relevant to our study. ***We have updated the related work section to emphasize it***, viewing it more as complementary than competing. While [4] provides valuable insights at the 2T-token scale, explores MLM-then-CLM training curricula, and evaluates generative tasks, our work focuses on a broader exploration of encoder-based training design choices (e.g., masking ratio, CLM-MLM share during pretraining) and provides an extensive evaluation across a wide range of representation benchmarks, with particular attention to compute efficiency and downstream stability.
>
> * **"The choice of 100B tokens is only focused on the fineweb edu data."**
>
> The diversity of domains in encoder data mixtures is indeed an important factor, and ***we now highlight this more explicitly in Section 8***. However, we chose to keep it fixed in this study, since it requires careful consideration and can act as a confounding variable on its own. Previous scaling-law work has taken the same approach, fixing both the data and the model architecture to keep compute requirements manageable [1, 5].
>
> We selected the FineWeb-Edu dataset because it provides an excellent efficiency-quality trade-off for large-scale pre-training of decoders [6] (to our knowledge, no similar dataset exists for encoders), making it a natural choice for our efficiency-focused 100B-token setup. Additionally, being fully open-source, the dataset facilitates easier replication of experiments.
>
> * **"The 75-25% scenario uses CLM on 75B tokens and MLM on 25B, which seems far from the more realistic scenario of a very large budget allocated to CLM."**
>
> The experiments mentioned refer to Section 4, where we combine CLM and MLM objectives sequentially with different ratios in a pretraining-from-scratch setup. The goal is not to start from an existing LLM already trained with CLM, typically on a very large number of tokens, but rather to provide insights on how a practitioner might allocate training steps given a fixed data budget.
>
> In Section 5, we explore a continued pretraining scenario, starting from an existing CLM model. We show that adapting the CLM model with 12k MLM steps achieves results comparable, on average, to MLM-only pretraining (Figure 9). Furthermore, as training progresses beyond 100B tokens (42k steps), the loss tends to plateau (***see Figure 12 in Appendix D, which we added in the updated paper version***), suggesting that starting from an already highly trained CLM model would likely require proportionally far fewer MLM steps [7]. We agree that confirming this intuition in a controlled setup would be a valuable direction for future work, but it is beyond the scope of our study in terms of compute.
>
> * **"It is hard to judge whether the [masking ratio] results are due to the relatively short training budgets."**
>
> We agree that 100B tokens may not be sufficient to fully maximize the performance of our encoders. However, existing work suggests that this scale is sufficient to produce interpretable signals [1, 3] and meaningful performance orderings between configurations, which aligns with the primary goal of our study. Evaluating an extensive set of pre-training configurations under substantially larger training budgets would have incurred a significant compute overhead.

---

> ### Author Response · Authors · 2025-11-26
>
> **Minor comments:**
>
> * **"Further analysis or explanations of differences in IR and token level tasks would be interesting."**
>
> This is a very interesting point. Intuitively, it may stem from the nature of token-level tasks, which rely less on bidirectionally informed context to identify named entities or word types. In contrast, tasks such as SC, IR, or QA depend more heavily on broader contextual understanding. While this remains an intuition that requires further verification, it already offers a practical takeaway for practitioners: for industry-oriented tasks like named entity recognition, the strongest backbone is not necessarily a pre-trained encoder, and decoder models can also perform very competitively. ***We have added a few words in Section 3 to emphasize this observation and the intuition behind it.***
>
> * **"Other analysis of the interaction between LR decay and curriculum would be interesting."**
>
> The interaction between learning rate decay and curriculum is indeed an interesting topic. However, for practical reasons, we relied on the Warmup-Stable-Decay (WSD) learning rate schedule, as it allowed us to save significant compute. Using a simple warmup-decay schedule would have required separate from-scratch pre-training runs for models trained on different numbers of tokens. In contrast, with WSD, it was possible to resume a longer run from the checkpoint of a shorter run, greatly improving efficiency.
>
> * **"The tables in the appendix [...] would benefit [...] bolding extreme values."**
>
> We thank the reviewer for this suggestion. ***We modified the tables in the appendix accordingly.***
>
> * **"Any further analysis or intuition on the u-shaped masking ratio?"**
>
> The impact of masking ratio in encoder pretraining has been an active area of research [8, 9, 10], To our knowledge, however, no prior work has fully disentangled its effect on task-specific downstream performance, highlighting that this is a complex problem requiring dedicated attention. It remains a promising avenue for further research, and ***we now mention it explicitly in Section 8.***
>
> We thank the reviewer once again for their feedback and hope our responses address their concerns. We remain available for any further clarification.
>
> **References:**
>
> [1] Training Compute-Optimal Large Language Models (Hoffman et al., 2022)
>
> [2] Beyond Chinchilla-Optimal: Accounting for Inference in Language Model Scaling Laws (Sardana et al., 2023)
>
> [3] BERT: Pre-training of Deep Bidirectional Transformers for Language Understanding, (Devlin et al., 2018)
>
> [4] Seq vs Seq: An Open Suite of Paired Encoders and Decoders (Weller et al., 2025)
>
> [5] Training Compute-Optimal Large Language Models (Hoffman et al., 2022)
>
> [6] The FineWeb Datasets: Decanting the Web for the Finest Text Data at Scale (Penedo et al., 2024)
>
> [7] LLM2Vec: Large Language Models Are Secretly Powerful Text Encoders (BehnamGhader et al., 2024)
>
> [8] Should You Mask 15% in Masked Language Modeling? (Wettig et al., EACL 2023)
>
> [9] Learning Better Masking for Better Language Model Pre‑training (Yang et al., ACL 2023)
>
> [10] Dynamic Masking Rate Schedules for MLM Pretraining (Ankner et al., EACL‑Short 2024)

---

### Official Review · Reviewer_JaBv · 2025-11-01

**Soundness:** 4
**Presentation:** 3
**Contribution:** 3
**Rating:** 8
**Confidence:** 3

**Summary:**

There has been much discussion in the community at the relative merits of Bert-style training for learning representations of text, in the age of next token prediction with large language models. This paper presents a careful empirical study as to  compare masked language modeling (MLM) and causal language modeling (CLM). They compare several models with less than or equal to 1B params, on several tasks. They find that MLM still generally beats out CLM, but training with CLM and then MLM generally outperforms any other alternative. They study continued pretraining where they find also that adapting a CLM trained model via MLM generally performs best.

**Strengths:**

1. Originality: Several studies have studied both MLM and CLM for learning representations of data, but to my knowledge none have such an in depth empirical study.

2. Quality: This paper uses strong state-of-the-art models as the baseline and evaluate on a wide array of tasks. As such, I consider the empirical results to be high quality

3. Clarity: The motivation and work done in this paper are quite clear.

4. Significance: The paper asks a clear and important question and answers is with good empirical evidence
This paper provides concrete recommendation to practitioners.

**Weaknesses:**

1. The scale is somewhat small, at 1B but is still within a reasonable range for such an experiment. It might be valuable to see how this looks for a 7B model.

2. It would be good to see performance on all the tasks in the MTEB (Massive Text Embedding Benchmark).

**Questions:**

1. Are there any experiments where we first train with MLM and then with CLM? Related, could one feasibly alternate between the two, e.g. per epoch or per batch?

---

> ### Author Response · Authors · 2025-11-26
>
> We thank the reviewer for their very positive feedback, highlighting the originality, clarity, and experimental setup of our work. We also appreciate their remarks, which we address below.
>
> * **"The [model] scale is somewhat small."**
>
> We chose to focus our study on smaller-scale models for several reasons. First, existing literature on pre-training encoder-only models predominantly examines architectures below the 1B parameter range [1, 2, 3, 4], which made this a natural choice for our work. More importantly, a comprehensive analysis of the key training design factors we identified, such as masking ratio and the CLM to MLM mix, was only feasible at smaller scales given our compute budget constraints.
>
> However, we fundamentally agree with the reviewer’s point. Although there is work on adapting decoders into encoders at the 7B scale [5], these studies are not conducted in fully controlled experimental settings, which makes this a promising direction for future research. ***We now mention this more explicitly in Section 8.***
>
> * **"It would be good to see performance on all the tasks in the MTEB."**
>
> It would indeed be interesting to evaluate performance on the full MTEB benchmark. However, achieving meaningful results on MTEB typically requires substantial contrastive post-training [6, 7, 8, 9, 10, 11], which we believe falls outside the scope of our study and would also introduce a significant compute overhead.
>
> Instead, we chose to cover a broad set of tasks specifically designed to evaluate pre-trained models directly, including sequence classification, token classification, extractive question answering, and information retrieval, with three datasets for each. We believe this provides a comprehensive view that is well suited for comparing the configurations under investigation.
>
> ***We have added a more explicit note on this point in Section 8.***
>
> * **"Are there any experiments where we first train with MLM and then with CLM?"**
>
> Such experiments are entirely valid, and we initially considered running them. However, results in Section 3 showed that standard MLM outperforms standard CLM, providing little justification for completing pre-training with CLM.
>
> Furthermore, recent studies supported this view, reporting that MLM-then-CLM training consistently leads to weaker performance on text representation tasks [12].
>
> * **"Could one feasibly alternate between the two?"**
>
> Alternating between CLM and MLM could be a very promising approach, which we admittedly did not consider when designing our experimental setup. However, this strategy may require the model to learn two distinct modes of operation and divide its capacity between them, with no guarantee that resources would be allocated effectively. In hindsight, we consider such an experiment somewhat uncertain in the context of the current study, but it remains an intriguing direction for future work. ***We now discuss this possibility more explicitly in Section 8.***
>
> We again thank the reviewer for their helpful feedback. We hope our responses resolve the issues raised and are happy to provide any additional clarification.
>
> **References:**
>
> [1] BERT: Pre-training of Deep Bidirectional Transformers for Language Understanding, (Devlin et al., 2018)
>
> [2] RoBERTa: A Robustly Optimized BERT Pretraining Approach (Liu et al., 2019)
>
> [3] DeBERTa: Decoding-enhanced BERT with Disentangled Attention (He et al., 2020)
>
> [4] Smarter, Better, Faster, Longer: A Modern Bidirectional Encoder for Fast, Memory Efficient, and Long Context Finetuning and Inference (Warner et al., 2024)
>
> [5] LLM2Vec: Large Language Models Are Secretly Powerful Text Encoders (BehnamGhader et al., 2024)
>
> [6] Approximate Nearest Neighbor Negative Contrastive Learning for Dense Text Retrieval (Xiong et al., 2020)
>
> [7] RocketQA: An Optimized Training Approach to Dense Passage Retrieval for Open-Domain Question Answering (Qu et al., 2021)
>
> [8] Text embeddings by weakly supervised contrastive pre-training (Wang et al., 2022)
>
> [9] Multitask contrastive learning for 8192-token bilingual text embeddings, (Mohr et al., 2024)
>
> [10] M3-Embedding: Multi-Lingual, Multi-Functionality, Multi-Granularity Text Embeddings Through Self-Knowledge Distillation (Chen et al., 2024)
>
> [11] Jina-embeddings-v3: Multilingual Embeddings With Task LoRA (Sturua et al., 2024)
>
> [12] Seq vs Seq: An Open Suite of Paired Encoders and Decoders (Weller et al., 2025)

---

### Official Review · Reviewer_9eov · 2025-11-01

**Soundness:** 3
**Presentation:** 3
**Contribution:** 2
**Rating:** 6
**Confidence:** 3

**Summary:**

The paper runs controlled ablations (same data, tokenizer, and encoder family; sizes 210M/610M/1B) to test whether encoder pretraining should remain MLM-only or benefit from a two-stage CLM→MLM recipe. The authors show the following findings (in my understanding) (1) MLM wins on average for representation tasks; (2) CLM is more data-efficient early in training and yields more stable fine-tuning; (3) a two-phase schedule that starts with CLM and then switches to MLM outperforms MLM-only at fixed compute; and (4) in continued pretraining, adapting a CLM with MLM beats continuing MLM on an MLM model—suggesting a practical path that leverages widely available decoder checkpoints.
Setup details: EuroBERT encoders, FineWeb-Edu English data, LLaMA-3 tokenizer; most two-stage/CPT studies use the 610M model and 40% masking for MLM.

**Strengths:**

- Thorough controls and scale of experiments. Same backbone family, shared data order, and systematic sweeps (sizes, masking ratios, schedules) reduce common confounders in MLM vs CLM comparisons. Compute and evaluation budgets are reported in detail.


- Actionable recipe: Under fixed compute, CLM→MLM (e.g., 25%→75%) consistently beats MLM-only; continued-pretraining results make a practical case for initializing from a decoder checkpoint, then adapting with MLM.


- Early data-efficiency + robustness: CLM learns useful representations faster early on and is less sensitive to fine-tuning learning rates; CLM→MLM is also less brittle to masking-ratio choices.


- Clarity on task sensitivity: The paper separates effects across SC/TC/QA/IR, noting where bidirectionality matters most (e.g., QA) and where CLM narrows the gap (e.g., IR), which is useful guidance for practitioners.

**Weaknesses:**

- General validity of the experiments might be narrow in scope: Results are tied to one encoder family and one tokenizer in English; generalization to other backbones (e.g., DeBERTa/ModernBERT), tokenizers, or languages is open.


- CPT evidence is concentrated at 610M and one masking ratio: While the authors sweep sizes and masks in base runs, most decisive experiments (schedules, CPT) are at 610M with 40% mask, which limits how confidently we can extrapolate.


- Magnitude of gains is modest: The CLM --> MLM uplift over strong MLM baselines appears small in many cases (Figure 2 scales across figures are a bit misleading in this regard, where each task have different scales)


- Scaling anomalies might need more explanation: Some configurations show 1B not strictly dominating 610M (e.g., QA at certain masking ratios), which makes me wonder if there was other training issues that could have contributed to this. It would be good if the authors provided some explanation on this.

**Questions:**

(covered above)

---

> ### Author Response · Authors · 2025-11-26
>
> We thank the reviewer for their comments, which we address below. We appreciate their positive feedback on our controlled training and evaluation setup, as well as the actionable insights we provide on pre-training encoder models, on data-efficiency and fine-tuning robustness.
>
> * **"General validity of the experiments might be narrow in scope."**
>
> The decision to focus on a single architecture, tokenizer, and language was intentional, ensuring a clearly defined experimental scope that remained as extensive as possible for analyzing our key factors of interest (masking ratio, the proportion of CLM steps in encoder training, continued pre-training experiments) while keeping computational costs manageable. Prior scaling-law studies follow the same strategy to avoid exponential growth in experimental budget [1, 2]. We also ensured that the fixed parameters we selected were standard, for example using the widely adopted Llama 3 tokenizer [16].
>
> For instance, running all MLM-CLM experiments with an additional tokenizer, even for the 610M model alone and without the masking-ratio experiments, would have required more than 20,000 extra GPU hours.
>
> Our choice to focus on English was also deliberate. Multilinguality introduces a separate research dimension, since the composition of the training mix can significantly affect outcomes and therefore deserves its own dedicated investigation [3, 4, 5].
>
> ***However, we agree these additional dimensions are important factors and have mentioned them more explicitly in Section 8.***
>
> * **"CPT evidence is concentrated at 610M and one masking ratio."**
>
> From a practicality standpoint, we had to make choices that kept computational costs reasonable.
>
> Regarding the masking ratio, as noted in Section 3 of our study, “there is no optimal masking ratio.” We therefore adopted 40%, which yielded a slightly better average performance across metrics, although differences between ratios were small.
>
> As for the model size, focusing on the 610M range was a deliberate compromise: most encoder-only architectures commonly used and downloaded on Hugging Face [6, 7, 8, 9] fall below the 1B parameter threshold due to efficiency requirements inherent to encoder models. Meanwhile, several strong decoder models also operate in the 300-700M range [10, 11], making 610M a representative and computationally practical choice for the core experiments.
>
> We also acknowledge that verifying results at a second scale is valuable. ***For this reason, we conducted an additional experiment at the 1B scale and reported in Appendix D (Figure 11) a comparison between pure MLM and hybrid CLM-MLM pre-training with a 25%-75% ratio, which confirms the consistency of our findings at a larger scale.***
>
> * **"The CLM → MLM uplift over strong MLM baselines appears small in many cases."**
>
> While the absolute magnitude of gains may sometimes appear modest because of the metric scales, we systematically report 95% confidence intervals based on fine-tuning with five different random seeds to provide scale-independent statistical evidence [5, 12, 13]. For example, in token classification, model differences often involve only a few challenging entities; nevertheless, these differences can still be statistically significant, as illustrated in Figures 2, 3, 6, 8, and 9.
>
> * **"Scaling anomalies might need more explanation."**
>
> We do not interpret these observations as anomalies, but rather as minor variations arising during pre-training. Although all models were pre-trained under identical conditions, interactions between design factors such as model size, masking ratio, and training curriculum cannot be fully controlled, especially within a single pre-training run.
>
> To account for this, we performed multiple fine-tuning runs to approximate the variance introduced by these pre-training configurations [6, 14, 15]. The results, shown with 95% confidence intervals (error bars in Figure 2, shaded areas in Figure 3), reveal no statistically significant anomalies. ***Following the reviewer’s suggestion, we have added a sentence in Section 3 to clarify this behavior.***
>
> In addition, it is not uncommon for smaller models to outperform larger ones on specific tasks. This is a frequent phenomenon and should not be considered anomalous as long as it remains localized to particular task types [16, 17, 18].
>
> We thank the reviewer once again for their thoughtful feedback. We hope that our responses have addressed their concerns, and remain available for any further clarification.

---

> > ### Author Response · Authors · 2025-11-26
> >
> > **References:**
> >
> > [1] Scaling Laws for Neural Language Models (Kaplan et al., 2020)
> >
> > [2] Training Compute-Optimal Large Language Models (Hoffman et al., 2022)
> >
> > [3] How Multilingual is Multilingual BERT? (Pires et al., 2019)
> >
> > [4] Unsupervised Cross-lingual Representation Learning at Scale (Conneau et al., 2019)
> >
> > [5] EuroBERT: Scaling Multilingual Encoders for European Languages (Boizard et al., 2025)
> >
> > [6] BERT: Pre-training of Deep Bidirectional Transformers for Language Understanding, (Devlin et al., 2018)
> >
> > [7] RoBERTa: A Robustly Optimized BERT Pretraining Approach (Liu et al., 2019)
> >
> > [8] DeBERTa: Decoding-enhanced BERT with Disentangled Attention (He et al., 2020)
> >
> > [9] Smarter, Better, Faster, Longer: A Modern Bidirectional Encoder for Fast, Memory Efficient, and Long Context Finetuning and Inference (Warner et al., 2024)
> >
> > [10] Qwen3 Technical Report (Bai et al., 2025)
> >
> > [11] SmolLM2: When Smol Goes Big – Data-Centric Training of a Small Language Model,  (Ben Allal et al., 2025)
> >
> > [12] What are the best systems? new perspectives on nlp benchmarking (Colombo et al., 2022)
> >
> > [13] Results of wmt23 metrics shared task: Metrics might be guilty but references are not innocent (Freitag et al., 2023)
> >
> > [14] Mixout: Effective Regularization to Finetune Large-scale Pretrained Language Models (Lee et al., 2019)
> >
> > [15] Fine‑Tuning Pretrained Language Models: Weight Initializations, Data Orders, and Early Stopping (Dodge et al., 2020)
> >
> > [16] The Llama 3 Herd of Models (Llama Team, 2025)
> >
> > [17] Qwen3 Technical Report (Qwen Team, 2025)
> >
> > [18] Inverse scaling can become U-shaped (Wei et al., 2022)

---

> > > ### Comment · Reviewer_9eov · 2025-11-27
> > >
> > > Thanks for addressing the questions and clarifying. I maintain my accept recommendation.

---

### Author Response · Authors · 2025-12-02
**Executive Summary to the AC**

We briefly summarize the reviews and answers to facilitate the AC’s assessment of our work.

**Strengths:**

The reviewers agree on the **strong relevance** of the work (*9eov*, *JaBv*, *e3tm*, *hZNQ*), emphasizing that it addresses an important and timely question for encoder design. They also highlight the **clarity of the presentation** (*9eov*, *JaBv*, *hZNQ*), describing the narrative as clear and easy to follow. The paper’s **extensive and large-scale** experimental exploration (*9eov*, *e3tm*, *hZNQ*) is repeatedly praised, particularly its investigation of mixed objectives (*e3tm*) and its insights into data efficiency and robustness (*9eov*). Finally, the reviewers emphasize the thorough and carefully **controlled setup** (*9eov*, *JaBv*, *e3tm*, *hZNQ*), whose systematic design and significance-aware evaluation result in reliable findings and actionable guidance for practitioners.

**Q&A:**

Reviewers raised questions primarily about the scope and scale of our experiments. We addressed these points through written clarifications and, where relevant, modified or complemented the paper with additional explanations or experiments. **All newly added content appears in blue in the new version**.

* **Q1**: Reviewer *9eov* asked about the narrow scope of the experiments, limited to a single encoder family, tokenizer, and language (English), and the concentration of key experiments at the 610M model scale.

$\rightarrow$ **Response:** We explain that focusing on a single architecture/language was a deliberate choice to ensure an extensive scope for analyzing our core factors (masking ratio, CLM-MLM ratio) while keeping compute manageable, a strategy common in large-scale controlled studies [1, 2]. We confirm the consistency of our findings at a larger 1B scale through an **additional experiment in Appendix D**.

* **Q2**: Reviewer *JaBv* suggested extending the evaluation benchmark to include MTEB tasks.

$\rightarrow$ **Response:** While we agree that these evaluations would further complement our analysis, extensive contrastive post-training would have introduced substantial compute overhead and could have added additional confounding factors. We therefore chose to focus on pre-trained models, for which we already provide a broad range of evaluation benchmarks. Nevertheless, in line with the reviewer’s suggestion, we explicitly **added a paragraph in Section 8 addressing this point**.

* **Q3**: Reviewer *e3tm* questioned the 100B token budget and the use of the FineWeb-Edu dataset, suggesting it might limit insights, especially for masking ratio analysis and practical data diversity.

$\rightarrow$ **Response:** We confirm that, although 100B tokens are insufficient for state-of-the-art performance, they are sufficient to observe meaningful and interpretable training dynamics for our comparative study [2]. We note that fixing the data mixture (FineWeb-Edu, known for providing efficient decoder training [3]) prevents it from becoming a confounding factor, following the practice of prior large-scale controlled studies [1, 2].

* **Q4**: Reviewer *hZNQ* requested that we complement our claim that CLM is more data-efficient than MLM by reporting FLOPs in addition to the data-sample-based analysis already provided.

$\rightarrow$ **Response:** **In Appendix D, we provide an additional figure** showing that CLM is more efficient than MLM not only in terms of data samples but also in terms of training FLOPs.

* Other remarks included questions on results significance (*9eov*), which we addressed thoroughly below by emphasizing the confidence intervals we report. Reviewers also asked about the task-specific impact of training design on downstream performance (*e3tm*, *hZNQ*), for example, trends in masking ratio depending on the task, or why CLM outperforms MLM on some tasks but not others. We acknowledge that these aspects could not be fully addressed in the current paper and have therefore deferred them to future work (Section 8).

**References:**

[1] Scaling Laws for Neural Language Models (Kaplan et al., 2020)

[2] Training Compute-Optimal Large Language Models (Hoffman et al., 2022)

[3] The FineWeb Datasets: Decanting the Web for the Finest Text Data at Scale (Penedo et al., 2024)

---

### Meta-Review · Area_Chair_6HwK · 2026-01-01

**Summary:**

This paper analyzes the impact of pretraining with MLM versus CLM on learning encoder representations by rigours controlled and comprehensive experiments. All reviewers agreed that this work addresses a relevant and interesting problem and it’s clearly written and logically organized. The thorough and carefully and  controlled ablations studies appreciated by all reviewers. Key findings and contributions include:

- CLM-trained models are more data-efficient and demonstrate improved fine-tuning stability compared to training with MLM.
- sequentially applies CLM and then MLM, achieves optimal performance under a fixed computational training budget.
- The proposed training strategy is promising when initializing from readily available pretrained CLM models.

**Reviewer Concerns:**

A common concern raised by all reviewers is the the scope and scale of our experiments. For instance, Reviewer 9eov worried about the genearliability, Reviewer JaBv suggested that explore 7B model is valuable, and Reviewer e3tm concerned the dataset scale. I found that the authors did a particularly good job of addressing their concerns (and thus mine) in the rebuttal and added the new content accordingly to their new version. I encourage the authors to include the additional experiment in their final version.

Another concern I have and also raised by Reviewer hZNQ is the lack of discussion on why certain pretraining paradigms align better with specific task types. I agree with the reviewer and encourage the author to discuss further (or even test) as it would  make the work more insightful and actionable for practitioners.

**Reviewer Scores:**

Reviewer e3tm or hZNQ might increase their Rating from 6 to 8 as I found that the authors did a particularly good job of addressing their concerns (and thus mine) in the rebuttal and added the new content accordingly to their new version.

---

### Decision · Program_Chairs · 2026-01-26

Accept (Poster)